# Landscape of allele-specific transcription factor binding in the human genome

Sergey Abramov [1,2,3,12], Alexandr Boytsov [1,2,3,12], Daria Bykova[4], Dmitry D. Penzar[1,2,3,4], Ivan Yevshin[5,6,7], Semyon K. Kolmykov[5,6,7], Marina V. Fridman[2], Alexander V. Favorov [2,8], Ilya E. Vorontsov [1,2], Eugene Baulin [3,9], Fedor Kolpakov [5,6,7], Vsevolod J. Makeev [2,3,10,11✉] & Ivan V. Kulakovskiy [1,2,11✉]

Sequence variants in gene regulatory regions alter gene expression and contribute to phenotypes of individual cells and the whole organism, including disease susceptibility and progression. Single-nucleotide variants in enhancers or promoters may affect gene transcription by altering transcription factor binding sites. Differential transcription factor binding in heterozygous genomic loci provides a natural source of information on such regulatory variants. We present a novel approach to call the allele-specific transcription factor binding events at single-nucleotide variants in ChIP-Seq data, taking into account the joint contribution of aneuploidy and local copy number variation, that is estimated directly from variant calls. We have conducted a meta-analysis of more than 7 thousand ChIP-Seq experiments and assembled the database of allele-specific binding events listing more than half a million entries at nearly 270 thousand single-nucleotide polymorphisms for several hundred human transcription factors and cell types. These polymorphisms are enriched for associations with phenotypes of medical relevance and often overlap eQTLs, making candidates for causality by linking variants with molecular mechanisms. Specifically, there is a special class of switching sites, where different transcription factors preferably bind alternative alleles, thus revealing allele-specific rewiring of molecular circuitry.

[1] Institute of Protein Research, Russian Academy of Sciences, Pushchino, Russia. [2] Vavilov Institute of General Genetics, Russian Academy of Sciences, Moscow, Russia. [3] Moscow Institute of Physics and Technology, Dolgoprudny, Russia. [4] Faculty of Bioengineering and Bioinformatics, Lomonosov Moscow State University, Moscow, Russia. [5] Federal Research Center for Information and Computational Technologies, Novosibirsk, Russia. [6] Sirius University of Science and Technology, Sochi, Russia. [7] BIOSOFT.RU LLC, Novosibirsk, Russia. [8] Johns Hopkins University School of Medicine, Baltimore, MD, USA. [9] Institute of Mathematical Problems of Biology RAS—The Branch of Keldysh Institute of Applied Mathematics of Russian Academy of Sciences, Pushchino, Russia. [10] State Research Institute of Genetics and Selection of Industrial Microorganisms of the National Research Center Kurchatov Institute, Moscow, Russia. [11] Engelhardt Institute of Molecular Biology, Russian Academy of Sciences, Moscow, Russia. [12] These authors contributed equally: Sergey Abramov, Alexandr Boytsov. ✉email: vsevolod.makeev@vigg.ru; ivan.kulakovskiy@gmail.com

Sequence variants located in noncoding genome regions attract an increasing researchers' attention due to the frequent association with various traits, including predisposition to diseases[1,2]. Single-nucleotide variants (SNVs) in gene regulatory regions may affect gene expression[3] by altering binding sites of transcription factors (TFs) in gene promoters and enhancers and, consequently, efficiency of transcription[4].

On the one hand, parallel reporter assays allow massive assessment of variants in terms of gene expression alteration[5,6] but do not reveal particular TFs involved. On the other hand, there are multiple ways to assess if a single-nucleotide substitution changes TF-binding affinity, from detailed measurements of the TF affinity landscape in vitro[7,8] to conventional experiments on individual sequence variants[9,10] and computational modeling[11–13]. However, it is not trivial to utilize these data for annotating SNV effects at the genome-wide scale in a cell type-specific manner.

The functional effect of single-nucleotide substitutions can be studied in heterozygous chromosome loci, where TFs differentially bind to sites in homologous chromosomes with alternative SNV alleles. Reliable evidence comes from modern in vivo methods based on chromatin immunoprecipitation followed by high-throughput sequencing (ChIP-Seq). ChIP-Seq provides a deep read coverage of TF-binding regions, and non-perfect alignments of reads often carry single-nucleotide mismatches arising from heterozygous sites. Statistical biases between the numbers of mapped reads containing alternative SNV alleles reveal the so-called allele-specific binding events[1,14] (ASB, Fig. 1a).

Chromatin accessibility often serves as a proxy for the regulatory activity of a genomic region[15]. Massive assessment of allele-specific chromatin accessibility in more than 100 cell types[16] reported more than 60 thousand of significantly imbalanced sites. Yet, so far, only 10–20 thousand ASBs were reported per study (Supplementary Table 1), and the potentially vast landscape of allele-specific TF binding remains mostly unexplored.

Reliable identification of ASBs (the ASB calling) requires high read coverage at potential sites, which results either from deep sequencing of individual ChIP-Seq libraries or from data aggregation across multiple experiments. Reprocessed ChIP-Seq data for hundreds of TFs and cell types are available in databases such as GTRD[17] and ReMap[18], opening a way to an integrative meta-analysis, which could yield raw statistical power to detect cell type- and TF-specific ASBs.

Straightforward meta-analysis of the ASBs has two major limitations. First, many ChIP-Seq data sets are obtained in aneuploid cell lines, and copy-number variants (CNVs) are common even for normal diploid cells. Both the chromosome multiplication and local CNVs affect the expected read coverage of the respective genomic regions[19] and bring about imbalanced read counts at SNVs, possibly generating false-positive ASB calls (Fig. 1a). There exist strategies to reduce this bias (Supplementary Table 2), in particular, the known CNV regions can be filtered out[20] or predicted from a computational analysis of the corresponding genomic DNA[21,22] (which is often used as the ChIP-Seq control sample) and incorporated in statistical criteria when evaluating the potential ASB calls[19]. However, in many published experiments, the input DNA data control was omitted in favor of other controls, such as preimmune IgG, or had a limited sequencing depth making it useless for CNV predictions. Furthermore, currently, there are no systematic data on global (chromosome duplications) and local (CNVs) structural variations across all cell types with public ChIP-Seq data on TFs. Even when the external data on structural variation are available for particular cells, it is not guaranteed that the same estimates would

be valid for ChIP-Seq data obtained elsewhere, since long-cultivated immortalized cell lines might keep accumulating unreported differences in genome dosage across chromosomes[23].

The second major problem in ASB calling is the so-called reference read mapping bias[21,24]. Standard read alignment tools generally map more reads to the alleles present in the reference genome assembly, as such mapping has lower or no mismatch penalties. To account for the reference read mapping bias, an ideal scenario involves mapping to individually reconstructed genomes[22,25] or computational simulations[20] that provide estimates of mapping probabilities to alternative alleles separately for each SNV (see Supplementary Table 2 for an overview). Yet, these solutions are not applicable to premade read alignments (which are usually obtained with a simple reference genome) and hardly applicable to understudied cell types or particular samples that do not provide enough data to reconstruct an individual genome.

In this work, we present a novel framework for ASB calling from existing read alignments or premade variant calls, accounting for the allelic dosage of aneuploidy and CNVs, and read mapping bias. With this framework, we have performed a comprehensive meta-analysis to identify ASBs in the human ChIP-Seq data from the GTRD database[17]. The database of Allelic Dosage-corrected Allele-Specific human Transcription factor binding sites (ADASTRA, http://adastra.autosome.ru) provides ASB events across 674 human TFs (including epigenetic factors) and 337 cell types. We demonstrate that the single-nucleotide polymorphisms (SNPs) with ASBs often overlap expression quantitative trait loci (eQTLs) and exhibit associations with various normal and pathologic traits. A comparison of data for multiple TFs highlights the cases where different TFs preferentially bind to different alleles, i.e., when a single-nucleotide substitution can change an entry point of the involved regulatory pathway. Finally, we discuss selected cases where the ASB at SNPs reveals molecular mechanisms of associations between SNPs and important medical phenotypes.

## Results

We present a reproducible workflow for ASB calling and meta-analysis across human TFs and cell types (Fig. 1b). First, the variants are called from premade ChIP-Seq read alignments against the hg38 genome assembly. Next, the variant calls are filtered by excluding homozygous and low-covered variants (<5 reads supporting any of two alleles), as well as variants absent from the dbSNP[26] common subset (as putative de novo point mutations). The filtered SNVs from related ChIP-Seq data sets (sharing the cell type and particular wet lab) are used to identify the cell type features (aneuploidy and CNVs). A total set of variants is used to assess the global read mapping bias that is used as the basis for statistical model parametrization. Finally, ASB calling is performed separately for each ChIP-Seq experiment, and the resulting allele read bias $P$ values are aggregated using the George–Mudholkar's method[27] for each SNV, either at the TF level (across ChIP-Seq data for a selected TF from all cell types) or the cell type level (across ChIP-Seq data for a selected cell type for all TFs).

We used the workflow to process 7669 ChIP-Seq read alignments from GTRD covering 1025 human TFs and 566 cell types, and detected more than 2 hundred thousand ASBs at more than 2 hundred thousand SNPs for various TFs and 3 hundred thousand ASBs for cell types passing the Benjamini–Hochberg (FDR) adjusted $P$ value of 0.05, see Fig. 1c, d for an overview. Reaching these numbers has become possible because of the large volume of the starting data (the filtered list of considered variant calls contained more than 54 million entries) and the advanced statistical framework that we describe below. An overview of the

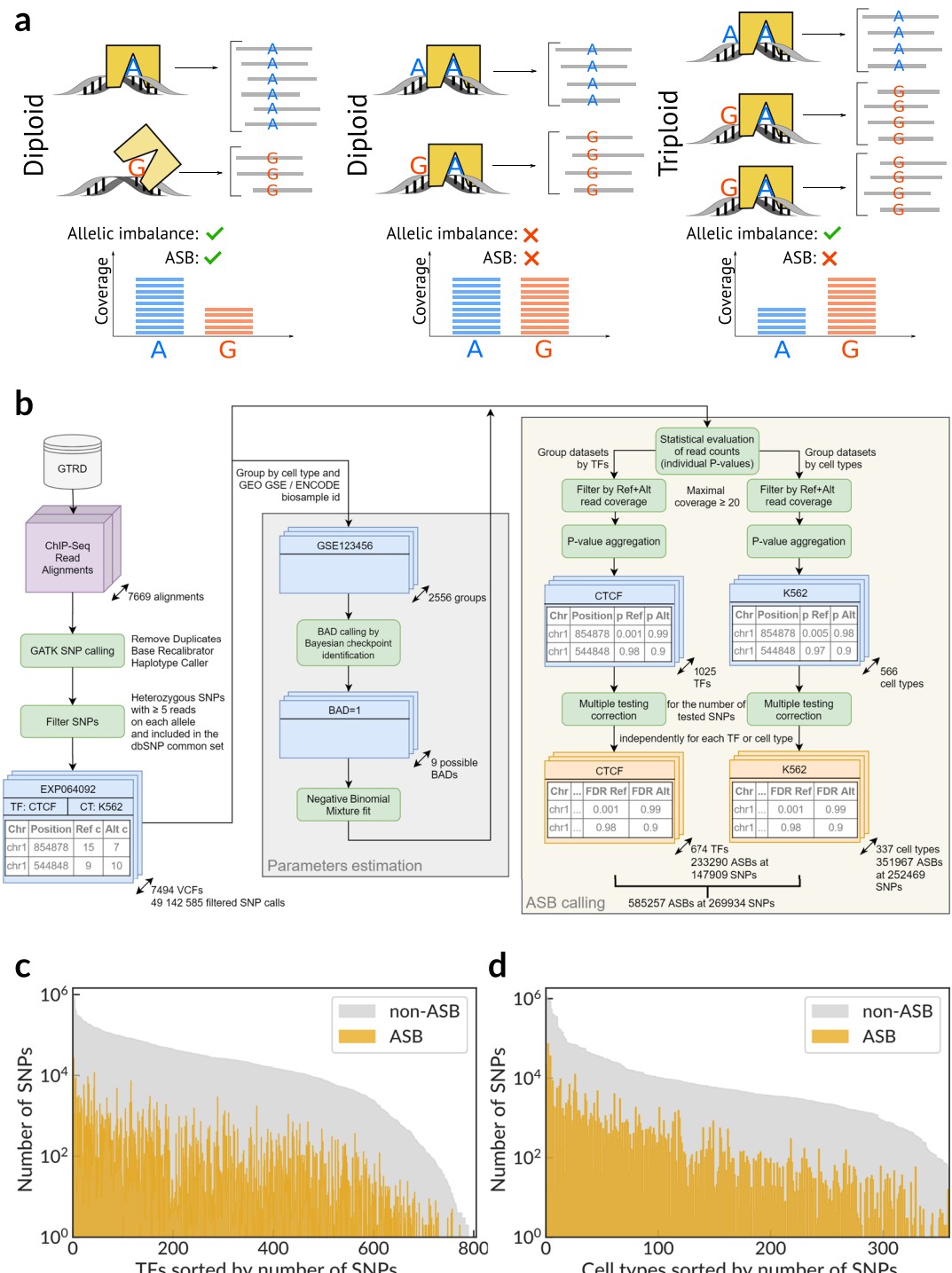

**Fig. 1 A scheme of allele-specific binding events, an overview of the ADASTRA pipeline, and its application to ChIP-Seq data. a** ChIP-Seq data allow detecting ASB events by estimating the imbalance of reads carrying alternative alleles. ASBs must be distinguished from sites where the allelic imbalance is caused by aneuploidy and copy-number variants. **b** The scheme of the ADASTRA pipeline: variant calling in read alignments from GTRD, estimation of statistical model parameters and background allelic dosage, filtering, and statistical evaluation of candidate ASBs. ADASTRA generates two complementary data sets: transcription factor ASBs (pairs of an SNP and a TF) and cell type ASBs (pairs of an SNP and a cell type). SNPs are annotated according to dbSNP IDs. **c, d** Number of SNPs (dbSNP IDs, Y-axis) with significant ASB events for various transcription factors and for various cell types. TFs or cell types (X-axis) are sorted by the number of SNPs. SNP single-nucleotide polymorphism, BAD background allelic dosage, ASB allele-specific binding, GTRD gene transcription regulation database, ADASTRA Allelic Dosage-corrected Allele-specific human Transcription factor binding sites.

processed data sets and variant calls per TF and cell type is shown in Supplementary Fig. 1.

**Estimating background allelic dosage (BAD) from single-nucleotide variant calls**. ASB is assessed against expected relative frequencies of reads supporting alternative alleles of a particular SNV in a particular genomic region. Assuming there was no read mapping bias, these expected frequencies would be mostly determined by the copy number of the respective genomic segments. In this study, we estimated the joint effect of local copy-number variation and global chromosome ploidy from the read counts at SNV calls, taking into account that the background for ASB calling is defined by the expected relative frequencies of the read counts supporting alternative alleles rather than by absolute allelic copy numbers.

We introduce BAD as the ratio of the major to minor allele dosage in the particular genomic segment, which depends on chromosome structural variants and aneuploidy. BAD can be estimated from the number of reads mapped at each allelic variant and does not require haplotype phasing. For example, if a particular genomic region has the same copy number of both alleles, e.g., 1:1 (diploid), 2:2, or 3:3, then it has BAD = 1, i.e., the expected ratio of reads mapped to alternative alleles on a heterozygous SNV is 1. All triploid regions have BAD = 2, and the expected allelic reads ratio is either 2 or ½. In general, if BAD of a particular region is known, then the expected frequencies of reads supporting alternative alleles are $1/(BAD + 1)$ and $BAD/(BAD + 1)$.

Importantly, accounting for BAD provides an answer to the question of the necessity of overdispersion in the statistical evaluation of ASBs[19,22]. In fact, a large portion of overdispersion of read counts disappears once the variant calls are segregated according to BADs of the respective genomic segments (see "Methods").

**BAD calling with Bayesian changepoint identification**. In this study, we present a novel method for reconstructing a genome-wide BAD map of a given cell type. The idea is to find genomic regions with approximately stable BAD using the read counts at SNV calls. Assuming that both differential chromatin accessibility and sequence-specific TF binding affect only a minor fraction of variants, the read counts for most of the SNVs must be close to equilibrium and thus provide imprecise but multiple measurements of BAD.

We have developed a Bayesian changepoint identification algorithm, which (1) segments the genomic sequence into regions of the constant BAD using dynamic programming to maximize the marginal likelihood and then (2) assigns BAD with the maximal posterior to each segment (see "Methods"). An additional preprocessing employs distances between neighboring SNVs to exclude long deletions and centromeric regions from BAD estimation. The BAD caller in action is illustrated in Fig. 2a for two chromosomes using ENCODE K562 data (see the segmentation map of the complete genome with multiple deletions in Supplementary Fig. 2).

We performed the BAD calling for 2556 groups of variant calls, where each group consisted of calls obtained from ChIP-Seq alignments for a particular cell type and GEO series or ENCODE biosample ID (i.e., for K562 cells of different studies, the BAD calling was performed independently). In BAD calling, recurrent SNVs sharing dbSNP IDs and found in different data sets within the same group were considered as independent observations. To systematically assess the reliability of the resulting BAD maps, we compared the predicted BADs at all SNVs with the ground truth BADs estimated from COSMIC[28] CNV data for 76 matched cell types, with K562 and MCF7 being the most represented. For K562 and multiple other cell types, the Kendall $\tau_b$ rank correlation was consistently better for joint data sets with higher numbers of SNVs (Fig. 2b), which justifies the usage of read counts at SNVs as point measurements of BAD.

Genome structural variations are the most likely yet not the only reason for unbalanced allelic dosage in a particular genomic region. In our case, the agreement of BAD and COSMIC copy-number maps confirms the validity of BAD estimates. However, even suboptimal agreement between a BAD map and the copy-number profile is not a problem as soon as the allelic dosage is estimated correctly.

Particularly, we found that BAD maps of MCF7 agreed poorly with COSMIC independently from the number of SNVs in the data set. To clarify the issue, we processed external deep genomic sequencing data for MCF7 with the ADASTRA pipeline (see "Methods"). The resulting BAD map from these data was not dependent on the ChIP procedure but agreed reasonably well with the MCF7 BAD maps from ChIP-enriched data sets, thus validating the ChIP-Seq-based BAD maps for MCF7 cells (see Supplementary Fig. 2).

Of note, the ChIP-independent BAD map for MCF7 still rather poorly agreed with the COSMIC copy numbers markup (SNP-level Kendall $\tau_b$ ~0.2), suggesting that for MCF7 the latter is likely an inadequate proxy for the actual BAD. We have no ultimate explanation for this observation but would like to remark that MCF7 was found among the most unstable cell types[29], which probably leads to discrepancies between exact CNV profiles and BAD estimates for cells originating from different studies.

Additionally, we have analyzed microarray-based CNV estimates for major cell types[29], including 13 cell types matching across these data, COSMIC, and our study (Supplementary Fig. 3). Interestingly, when compared to COSMIC, those data showed a higher correlation for MCF7 rather than for K562 cells. To a varying degree, such discrepancies can be observed for other cell types. Thus, careless recruitment of copy-number profiles obtained with different methods from different data sources as estimates of BAD may reduce the reliability of called ASBs, the disadvantage that is avoided by using BAD estimates directly from ChIP-Seq data.

As a separate test, we used the predicted BAD maps as multiple binary classifiers for different BAD values using SNP calls across all cell types. With the COSMIC data as the ground truth, we plotted a receiver operating characteristic (ROC) and a precision-recall curve (PRC) for each BAD (Fig. 2c, d). For the most widespread BADs (1–3) covering more than 90% of candidate SNVs (Supplementary Fig. 3), we reached >0.83 area under curve for ROC and 0.66–85 for PRC (Supplementary Table 3), proving the reliability of the predicted BAD maps.

With BAD maps at hand, we segregated the variant calls from all data sets by BAD and by fixed read coverage either at reference or alternative alleles. Then, for each such set of SNVs, we fitted the background distribution as a mixture of two negative binomial distributions with BAD-determined $p$ parameters (see "Methods"). ASBs were called independently for the reference (Ref-ASB) and the alternative (Alt-ASB) allele using separately fit background distributions for the fixed read counts at alternative and reference alleles, respectively, thus accounting for general read mapping bias.

**Overview of the ADASTRA database**. The results of the ASB calling are provided in the ADASTRA database (the database of ADASTRA factor binding sites). In ADASTRA, each dbSNP ID can have several ASB entries for different TFs or cell types. ADASTRA consists of two parts: the first part (TF-ASB, 233290

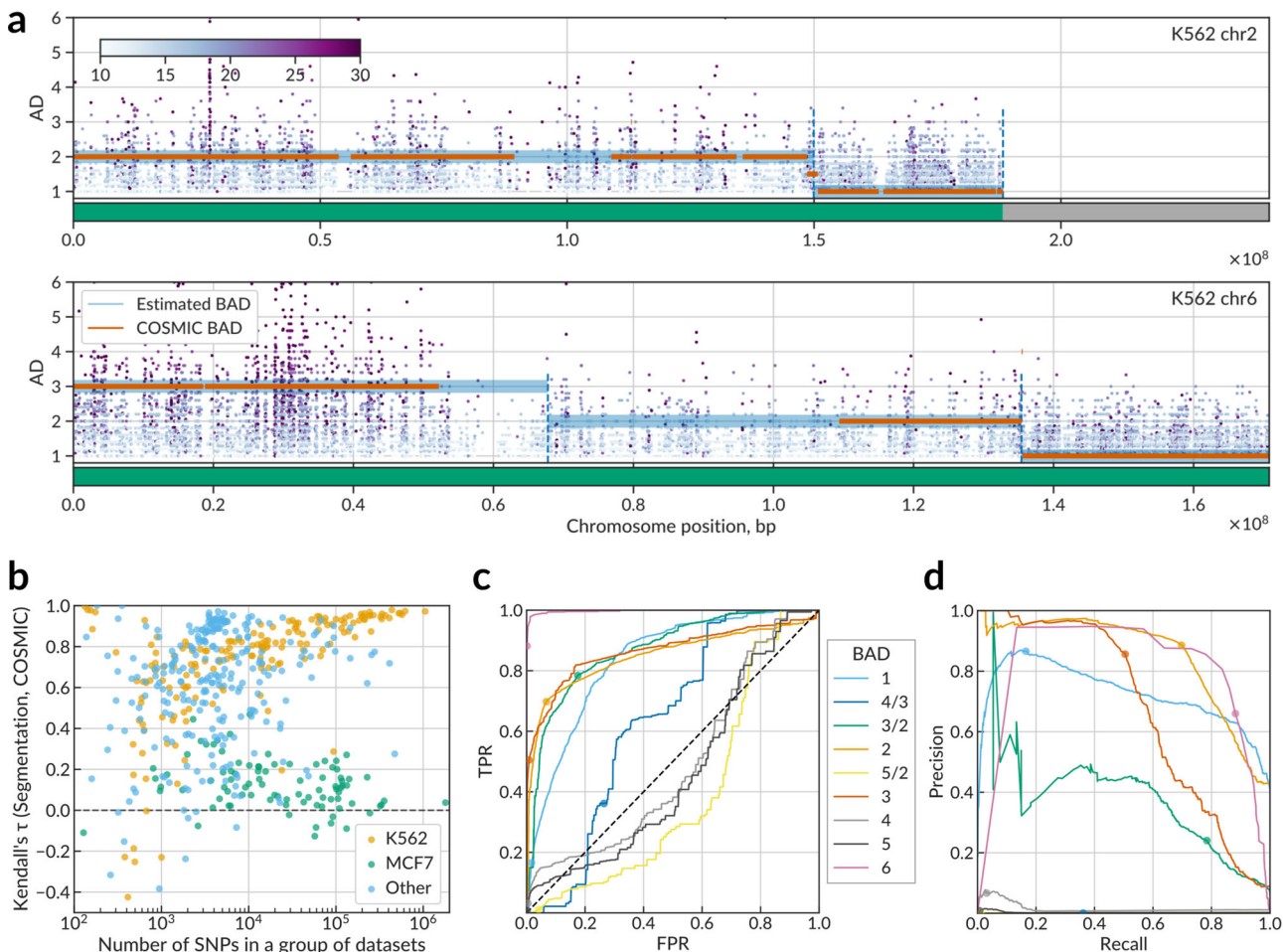

**Fig. 2 Bayesian changepoint identification allows reconstructing reliable genome-wide maps of background allelic dosage from single-nucleotide variant calls. a** BAD calling with Bayesian changepoint identification applied to variant calls detected at chr2 and chr6 in K562 ENCODE data (ENCBS725WFV). *X*-axis: chromosome position, bp. *Y*-axis: the allelic imbalance of individual SNVs. Horizontal green lines (ground level of the plots) indicate results of the initial stage of the algorithm: the detection of SNV-free regions including deletions, telomeric, and centromeric segments. Horizontal light-blue lines: predicted BAD. Orange dashes: "ground truth" BAD according to the COSMIC data (when available). **b** *Y*-axis: SNV-level Kendall $\tau_b$ rank correlation between the predicted BAD and the "ground truth" BAD (COSMIC data). Each of 516 points denotes a particular group of related data sets of the same series (ENCODE biosample or GEO GSE ID) and the same cell type. *X*-axis: the number of SNV calls in a particular group of related data sets. Only SNVs falling into regions of known BAD (present in the COSMIC data) are considered, recurrent SNVs in several data sets are considered only once. **c**, **d** Receiver operating characteristic and precision-recall curves for predicted BAD maps used as binary classifiers of individual SNVs according to BAD vs the "ground truth" COSMIC data. To plot each curve, the score $S = L(\text{BAD} = x) - \max_{y \neq x} L(\text{BAD} = y)$, where $L$ denotes log-likelihood, was used as the prediction score for thresholding. Colored circles denote the values obtained with the final BAD maps where particular BAD values were assigned to each segment according to the maximum posterior. Regions with BAD of 1, 3/2, 2, and 3 contain more than 97% of all candidate ASB variants. SNP single-nucleotide polymorphism, SNV single-nucleotide variant, AD allelic dosage, BAD background allelic dosage, TPR true positive rate, FPR false positive rate.

ASBs at 147909 SNPs) contains ASB obtained by aggregation of individual *P* values for each TF over cell types. The listed ASBs passed multiple testing correction ($P < 0.05$ after Benjamini–Hochberg adjustment for the number of tested ASBs). *P* value estimation (see below), aggregation, and multiple testing correction were performed separately for ASBs with preferred binding to the reference (Ref-ASB) and alternative (Alt-ASB) alleles, and for each TF. The other part of the database (CT-ASB, 351967 ASBs at 252469 SNPs) contains a similar aggregation of individual ASBs over TFs for each cell type.

TFs and cell types were unequally represented in the source data. Thus, the numbers of the resulting ASB calls were also biased toward most studied cell types and TFs (Fig. 3a, b), with the top contributions from CTCF for TFs and K562 for cell types. However, the top 8 TFs and top 5 cell types covered only half of ASB calls (for cell types) or less than a half of ASB calls (for TFs);

thus, the produced data on ASB events is diverse across different samples.

Next, we assessed how ASBs and candidate SNVs are distributed in different genomic regions (Fig. 3c). Compared to all SNVs and tested candidate ASB sites, the significant ASBs were enriched in enhancers (~4x more than expected from the number of SNVs for which there were candidate ASBs, Fisher's exact test $P < 10^{-300}$) and promoters (~3x more than expected, $P < 10^{-300}$). We consider this observation consistent with both the actual location of functional TF-binding sites and deeper coverage of the actual TF-binding regions with ChIP-Seq reads. In fact, ASBs are likely to cluster at the scale of the typical ChIP-Seq peak width, as revealed by the distribution of pairwise distances between SNVs with and without ASBs, which has a bimodal shape (Supplementary Fig. 4). This effect is likely caused by peak-scale clustering of ChIP-Seq reads allowing for higher

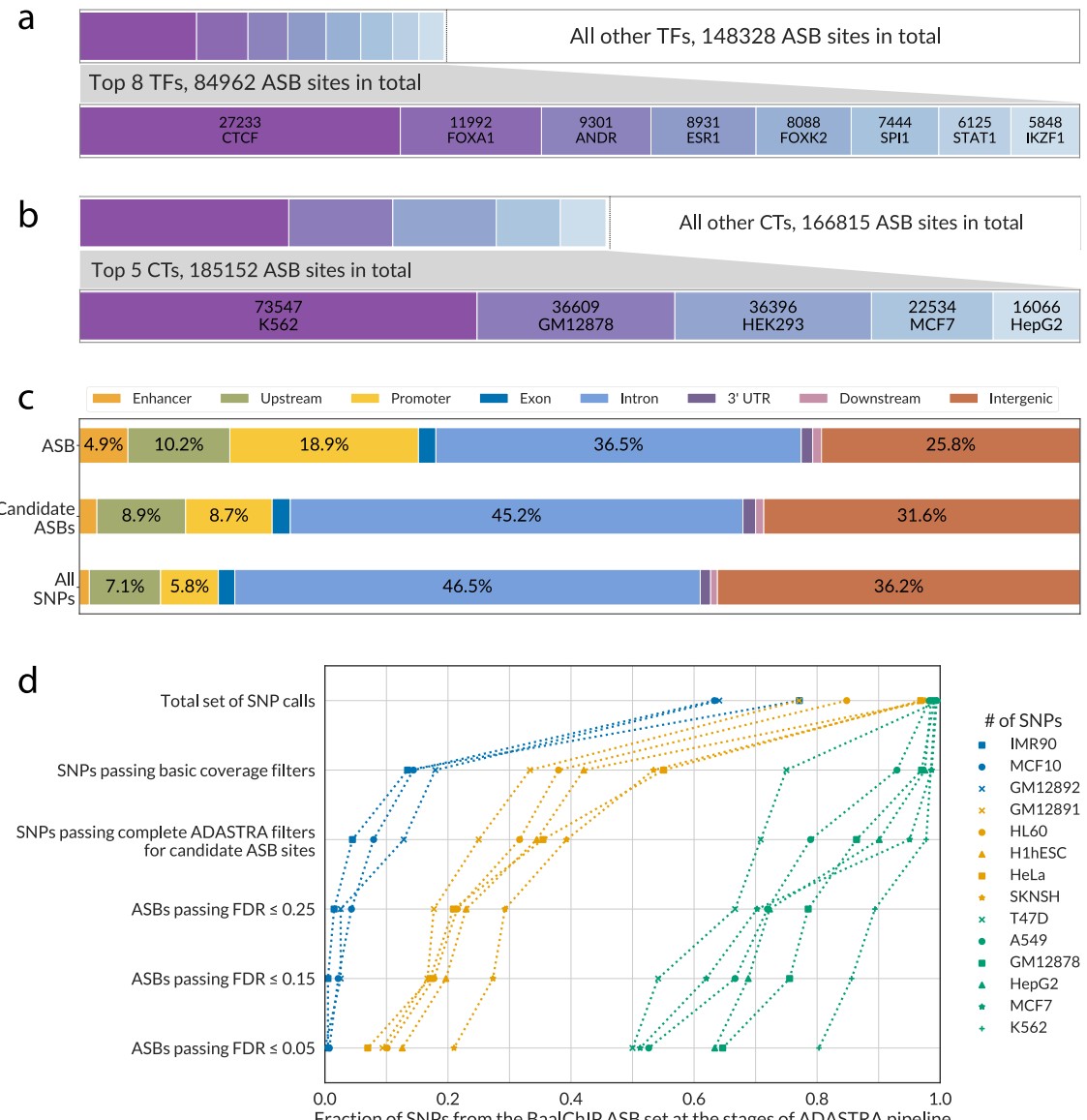

**Fig. 3 An overview of the ADASTRA ASBs and their genomic localization. a**, **b** The distribution of ASBs across TFs and cell types is not uniform. The top 8 TFs and top 5 cell types provide only nearly one third (TFs) or one half (cell types) of significant events. The bottom bars in each pair show the zoomed-in data for the top 8 TFs and top 5 cell types sorted by descending number of ASBs. **c** The complete bars correspond to the full set of SNPs (unique dbSNP IDs) with significant ASBs. The ASBs are more often found in promoters and enhancers as compared to either SNVs with candidate ASBs or all detected SNVs. The percentage of ASB-carrying SNPs falling into particular types of genomic regions is shown on bar labels. Top bar: significant ASBs (passing 5% FDR, 269,934 sites in total); middle bar: SNPs with candidate ASBs (passing the coverage thresholds and tested for significance, 2,024,836 sites in total); bottom bar: all SNPs detected in the variant calling (4,976,303 sites in total). **d** The fraction of BaalChIP-reported SNPs (*X*-axis) with allele-specific binding passing the filters at various stages of the ADASTRA pipeline (*Y*-axis). We considered data from 14 cell lines matching between BaalChIP ASB set and ADASTRA (with the ADASTRA ASBs reaggregated considering only 316 data sets shared between BaalChIP and ADASTRA out of a total of 548 BaalChIP ChIP-Seq data sets). The following checkpoints of the ADASTRA pipeline were considered: 1 Total set of SNP calls: SNPs found by GATK; 2 SNPs passing basic coverage filter: SNPs with ≥5 reads supporting each of alternative alleles; 3 SNPs passing complete ADASTRA filters for candidate ASB sites: heterozygous dbSNP common SNPs with total coverage of at least 20 reads in at least one experiment located in a chromosome eligible for BAD estimation, i.e., with ≥100 SNP calls at stage 2; 4 ASBs passing a fixed FDR: cell type level aggregated ASBs passing a given FDR threshold (Benjamini–Hochberg-corrected *P* value allowing for BAD). ASB *P* values were estimated by logit aggregation of the one-tail Negative Binomial *P* values across the experiments (see "Methods") and then the FDRs were estimated with Benjamini–Hochberg procedure. CT cell type, TF transcription factor, SNP single-nucleotide polymorphism, ASB allele-specific binding, FDR false discovery rate.

sensitivity of both SNP calling and ASB calling in the vicinity of ChIP-Seq peak summits.

We also compared the SNPs listed in ADASTRA with those of the previous ASB collections (Supplementary Fig. 5). ADASTRA includes ASBs at 38%, 44%, 57%, and 64% of dbSNP SNPs reported as ASBs in AlleleDB[22], and collections published in[20,30],

and[19], respectively. We additionally assembled a reproducible ASB set consisting of 2039 SNPs with ASBs detected in any two of those four ASB sets and found that ADASTRA included 1573 (77%) of the respective SNPs. Of note, taking pairwise, these four existing ASB data sets also poorly overlap each other (see Supplementary Table 4), suggesting that the major fraction of

ASBs is non-reproducible between studies and arise either from particular ChIP-Seq data sets or from unique procedures of different ASB calling pipelines.

To study in detail why ADASTRA failed to capture ASBs found in other studies, we used the set of ASB SNPs identified by one of the most advanced methods for ASB calling, BaalChIP[19]. ASB event could be missed at the SNP calling stage, could fail to pass the read coverage thresholds, or fail to pass the significance threshold for FDR-corrected $P$ value estimated against BAD. To assess the contribution of different stages of our pipeline to ASB calling sensitivity, we performed a stage-by-stage analysis of the underlying SNP set (see Fig. 3d). It turned out that the fraction of BaalChIP ASB SNPs recovered by ADASTRA was different for different cell types, with most of ASBs recovered for the cell types with the deepest sequencing coverage.

On the one hand, the basic coverage filters significantly reduced the number of SNPs under consideration resulting in a major loss in the fraction of recovered ASBs. On the other hand, we did not observe critical effects from any of the subsequent stages. For all cell types, the number of BaalChIP ASB SNPs recovered by ADASTRA decreased monotonously, suggesting that there was no particular bottleneck defining the sensitivity of the whole pipeline except the basic coverage filters. As more sites were recovered for cell types with better coverage, one can predict that the difference between different ASB calling pipelines would decrease as soon as more ChIP-Seq data would become available for analysis.

In general, there is an overlap between ADASTRA ASBs and the existing data on regulatory SNPs, including sites of allele-specific DNA accessibility[16] and reporter assay quantitative trait loci[6] (Supplementary Fig. 5), but the vast majority of ADASTRA data are novel.

Given the diversity of assessed TFs, it became possible to systematically compare SNVs carrying TF ASBs and identify the pairs of TFs preferring to share ASBs (Supplementary Fig. 6). Indeed, hundreds of TF pairs are significantly enriched for common ASBs (one-tailed Fisher's exact test $P$ value from 0.05 to $10^{-300}$ upon correction for multiple tested TF pairs with the Benjamini–Hochberg procedure). As a rule, shared ASBs were not related to interacting TFs (considering protein–protein interactions from STRING-db[31]). However, there was a systematic overlap between ASBs for chromatin-interacting epigenetic factors and related proteins, suggesting many of shared events are "passengers" in regions of allele-specific chromatin accessibility with TFs bound only to the accessible chromosome. Still, some interacting proteins (such as CTCF-RAD21) strongly prefer to share ASBs, and the same holds for particular composite elements of binding sites such as AR-FOXA1[32].

**Motif annotation is concordant with ASB calls**. For TFs specifically interacting with DNA, it is possible to perform computational annotation of ASBs with TF-recognized sequence motifs[33]. When a strong binding site overlaps an ASB SNP and the alternating alleles directly change the key nucleotides in the TF-binding DNA sequence, this SNP likely relates to different TF-binding affinity to the sites at homologous chromosomes, which directly produce the ChIP-Seq allelic imbalance. We call such events "driver" ASBs to distinguish them from side effects of piggyback TF binding and chromosome-specific local chromatin accessibility, the examples of "passenger" ASBs. Motif annotation highlights the driver ASBs and allows comparing the observed ASB effect (the allelic imbalance) and the effect predicted by sequence analysis (the difference in binding specificity reflected in the motif prediction scores), providing an independent evaluation of the reliability of ASB calls.

An ASB was considered as overlapping the TF motif occurrence if the TF position weight matrix (PWM) scored a hit with $P \leq 0.0005$ for any of the two alleles. The log ratio of $P$ values corresponding to PWM hits at alternative alleles was used as an approximation of the TF affinity fold change (FC). Fig. 4a compares the ASB significance ($X$-axis, signed $\log_{10}$ FDR; the sign set positive for Alt-ASBs and negative for Ref-ASBs) with the log ratio of motif hits $P$ values ($Y$-axis) for 218 TFs having at least 1 ASB within a motif hit. Predominantly, at heterozygous sites, alleles with more specific motif hits are covered with more ChIP-Seq reads, revealing the prevalence of motif-concordant ASB events (blue dots in Fig. 4a). Such concordance persists for more than 80% of SNVs with ASB allelic imbalance FDR < 5%, growing with decreasing ASB FDR and saturating at about 90% of SNVs (Fig. 4b). At 5% FDR, good motif concordance stands for many TFs, as illustrated by the top 10 TFs with the highest number of motif hits at ASBs (Fig. 4c). Importantly, even at larger FDR, there are more concordant than discordant ASBs.

Yet, for ~10–20% of SNVs, the motif hit odds ratios are discordant with the allelic imbalance (corrected for BAD), that is, more reads are attracted to the weaker motif hit (red dots in Fig. 4a and red bars in Fig. 4c). We believe that in such cases the allelic imbalance arises from other contributors (allele-specific chromatin accessibility or indirect TF binding), which override the sequence-specific TF affinity. Also, we use the motif prediction scores as a proxy of the TF-binding affinity and it is possible that the observed limited discordance partly reflects the imperfectness of the utilized motif models.

To quantify ASB allelic imbalance for BAD other than one, we defined the ASB effect size (ES) as follows (see "Methods" for details). For individual SNV (SNV in a single data set):

$$ES_{Ref} = \log_2\big(C_{Ref}/E\big(C_{Ref}|C_{Alt}\big)\big) \text{ and}$$
$$ES_{Alt} = \log_2\big(C_{Alt}/E\big(C_{Alt}|C_{Ref}\big)\big)$$

Here $C_{Ref}$ and $C_{Alt}$ are the read counts at the Ref and Alt alleles, and $E$ is the expectation. For BAD = 1: $ES_{Ref} \approx \log_2(C_{Ref}/C_{Alt})$.

The aggregated ES of an ASB is calculated as a weighted mean of ES values for the same allele for SNVs aggregated at the same genome position over TFs or cell types, with weights equal to negative logarithms of individual $P$ values, separately for each of the alleles.

BAD-corrected estimates of the ASB ES allow to visualize the magnitude of allelic imbalance at different positions of significant motif hits. To this purpose, we introduce a staveplot (Fig. 4d and Supplementary Fig. 7) that is partitioned into sections corresponding to the motif positions, and each section is a stave of four strings denoting the minor allele. Individual ASBs are shown as the beads on the staves, with the major allele encoded with color, following the palette of the motif logo diagram that is shown underneath. As an illustrative example, we use ASBs of the CEBPB TF (Fig. 4d). For example, the first string from the left denotes A as the minor allele found in the first position of CEBPB motif hits. The string carries multiple beads, each of which is the major allele of a particular heterozygous SNV within an ASB site of CEBPB. The position of a bead on the $Y$-axis shows the ASB ES in log-scale. The most conserved motif positions 3-7-9-10 are almost unicolor, with the major allele usually being the same as the consensus letter in the motif (hence the beads on the strings depicting minor alleles mostly share the color of the preferred major allele). Lowly conserved positions (e.g., 1 or 12) allow for more options with various pairwise combinations of alleles (i.e., with minor allele strings carrying the beads of all four possible colors). Of note, the staveplot reveals a clear pattern where beads found in core motif positions are located generally higher, i.e., marking a greater ES for heterozygous variants at conserved motif

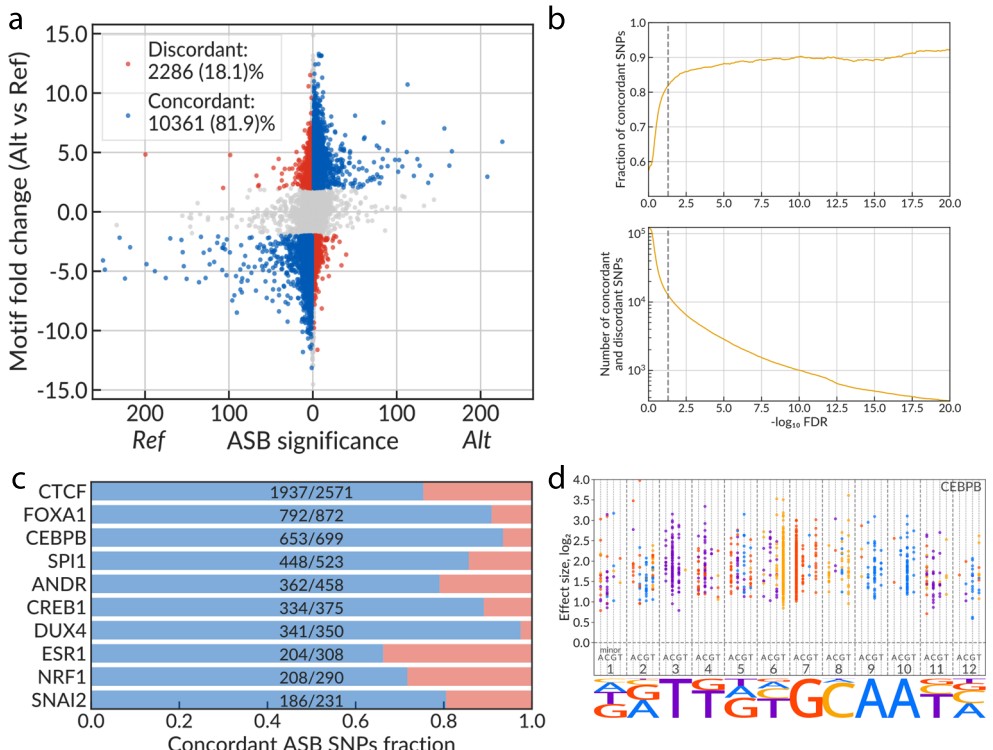

**Fig. 4 Motif annotation of SNPs agrees with TF-ASB calls. a** Scatterplot of the motif fold change (the predicted change in TF-binding affinity) vs the ASB significance for TFs that have PWMs in HOCOMOCO v11 core collection. The plot shows only the ASBs that overlap the TF motif occurrence (TF motif PWM hit with $P$ value ≤ 0.0005). $X$-axis: signed ASB significance, the absolute value is max($-\log_{10}$ FDR Ref-ASB, $-\log_{10}$ FDR Alt-ASB). The sign is set to negative if Ref-ASB is more significant than Alt-ASB (positive otherwise). $Y$-axis: motif fold change (FC) estimated as the $\log_2$-ratio of motif PWM hit $P$ values between the reference and the alternative alleles (the positive value corresponds to a higher affinity to alternative allele). The SNPs are marked as concordant (discordant) and colored in blue (red) if they exhibit significant ASBs (FDR ≤ 0.05), have motif |FC| ≥ 2, and the preferred allele of the ASB corresponds to (is opposite to) that of the TF motif. ASB $P$ values were estimated by logit aggregation of the one-tail Negative Binomial $P$ values across the experiments (see "Methods") and then the FDRs were estimated with Benjamini–Hochberg procedure. **b** The total number of discordant and concordant SNPs and the fraction of concordant SNPs among them ($Y$-axis) depending on the ASB significance cutoff, $-\log_{10}$ FDR ($X$-axis). **c** Barplot illustrating the proportion of SNVs with concordant and discordant ASBs for top 10 TFs with the largest total numbers of eligible SNVs. **d** The staveplot illustrating motif analysis of significant CEBPB ASBs. Each bead represents an SNV that is ASB and overlaps the predicted CEBPB binding site ($P$ value ≤ 0.0005) and has motif |fold change| ≥ 2. The X-coordinate shows the SNV position in the motif (underlined by the motif logo), the individual dashed strings denote four possible minor alleles at each position, the bead color is defined by the major allele. The strand orientation of ASBs is aligned to the predicted motif hits. $Y$-axis shows the ASB effect size. SNP single-nucleotide polymorphism, ASB allele-specific binding, PWM position weight matrix.

positions. This agrees with the commonly accepted testimony that substitutions in the core motif positions bring about larger changes in TF-binding affinity.

Particularly for CEBPB, position 6 is of special interest: it displays frequent T/C ASBs with C being the major allele. These cytosines belong to the core CG pair which is prone to spontaneous deamination. The produced mismatches are then protected from repair through enhanced CEBPB binding resulting in mutation fixation[34]. Such ASBs, on the one hand, confirm frequent mutagenesis of CEBPB binding sites, and, on the other hand, suggest the action of purifying selection that stabilizes such sites as heterozygous variants. The staveplots for other TFs are shown in Supplementary Fig. 7.

**Machine learning predicts ASBs from sequence analysis and chromatin accessibility.** With previously published ASB sets of smaller volumes, it was possible to predict ASB from chromatin properties and a sequence analysis[20]. To assess to what degree this holds for ADASTRA data, we applied machine learning with a random forest model[35] atop experimentally determined allele-specific chromatin DNase accessibility data[16], predicted allele-specific chromatin profile from DeepSEA[11], and sequence motif hits (Supplementary Table 5).

A generic classification problem (ASBs vs non-ASBs) can be formalized in two subtasks: (1) general assessment, i.e., to predict if an SNV makes the ASB for any of the TFs or in any of the cell types, and (2) TF- and cell type-specific assessment, i.e., to predict if an SNV makes the ASB for the particular TF or in the particular cell type. Models for both subtasks were trained and validated using multiple single-chromosome hold-outs: iteratively for each of 22 autosomes, one autosome was selected for validation, and 21 other autosomes were used for training. At each iteration, the model performance was estimated at the held-out autosome, and the resulting ROC and PRC were averaged.

For the first subtask, the performance at TF and cell type ASBs was 0.74 and 0.73 for the area under the receiver operating characteristic (auROC), and 0.44 and 0.56 for the area under the precision-recall curve (auPRC), respectively (see the plots in Supplementary Fig. 8). For the second subtask, we used the top 10 TFs and top 10 cell types with the highest numbers of ASBs, and a dedicated model was trained for each TF and each cell type (Supplementary Table 6 and Supplementary Fig. 8). The quality of the models was different for different TFs and cell types, with the highest auROC of 0.72 and 0.81 for CTCF (of TFs) and HepG2 (of cell types), and the highest auPRC of 0.35 and 0.64 for CTCF and A549. Of note, RAD21 ASBs were also predicted with

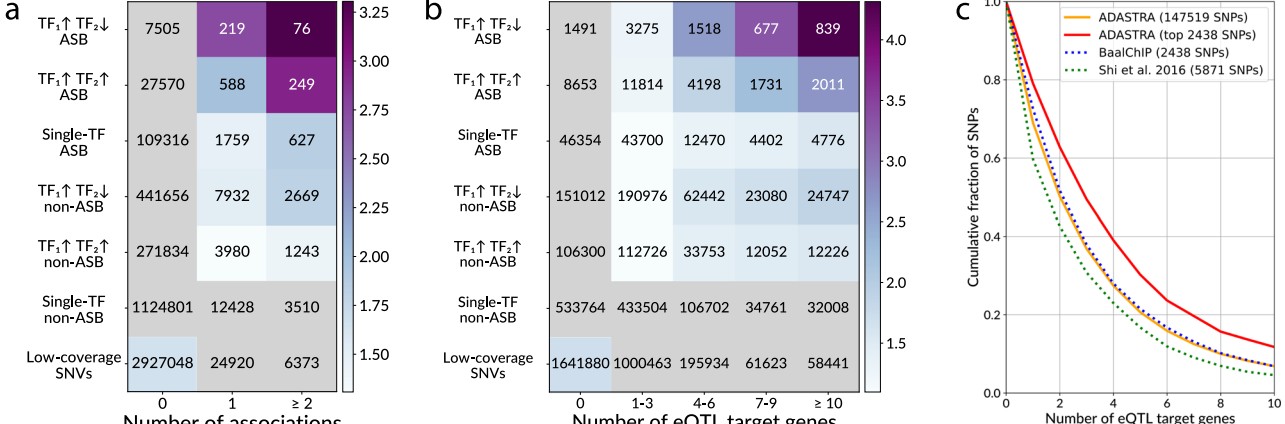

**Fig. 5 ASBs are enriched with pathologic phenotype associations and eQTLs. a, b** Enrichment of ASBs among phenotype-associated and eQTL SNVs. *Y*-axis denotes several exclusive groups of SNPs: TF1↑TF2↓, SNVs carrying both Ref- and Alt-ASBs of different TFs, i.e., where at least two TFs prefer to bind alternating alleles; TF1↑TF2↑, SNVs carrying ASBs for at least two TFs preferring to bind the same allele; single TF, SNVs with ASB of a single TF; low-covered SNVs that did not pass a total coverage threshold ≥ 20. Non-ASBs are SNVs with the TF-ASB FDR > 0.05. *X*-axis: **a** the number of unique (dbSNP ID, trait, database) triples for a given SNV considering four databases of SNP-phenotype associations (EBI, ClinVar, PheWAS, and BROAD autoimmune diseases fine-mapping catalog); **b** the number of eQTL target genes according to GTEx eQTL data. The coloring denotes the odds ratios of the one-tailed Fisher's exact test for the enrichment of SNVs with associations for each group of ASBs (against all other SNVs in the table). The gray cells correspond to nonsignificant enrichments with *P* > 0.05 after Bonferroni correction for the total number of cells. The values in the cells denote the numbers of SNVs. **c** The fraction of ASB SNPs from particular ASB collections (*Y*-axis) coinciding with GTEx eQTLs passing a certain threshold for the number of target genes (*X*-axis). Fourteen cell types overlapping between ADASTRA (solid orange line) and BaalChIP (dotted blue line) data have been considered: A549, GM12878, GM12891, GM12892, H1hESC, HL60, HeLa, HepG2, IMR90, K562, MCF10, MCF7, SKNSH, T47D. Data for HeLa and GM12878 cells were extracted from the Shi et al. collection (dotted green line). A subset of 2438 top-significant ADASTRA ASBs (the subset size equal to that of the BaalChIP set) is additionally shown to illustrate that these ASBs relatively more often coincide with potent eQTLs. SNP single-nucleotide polymorphism, SNV single-nucleotide variant, ASB allele-specific binding, eQTL expression quantitative trait loci.

very high reliability, as they are often located at the same variants as CTCF ASB.

Analysis of the feature importance (Supplementary Fig. 8) showed that all models utilized signals from the final layer of the DeepSEA neural network that was specifically designed to distinguish regulatory SNVs. Of note, among multiple DeepSEA features, those for the matched cell types were automatically prioritized. In agreement with previous studies[16,20], the models also obtained useful information from the experimental DNase-Seq data, and the data on allelic imbalance were generally more important than the basic read coverage. In the case of ASBs of particular TFs, motif-based features further facilitated distinguishing ASBs from non-ASBs. We expect that the same framework can allow further improvement of ASB prediction when supplied with additional chromatin accessibility and allelic imbalance data from matched cell types and with improved models of TF-binding sites.

**Disease-associated SNPs and eQTLs are enriched with ASBs.** To assess if ASB facilitates the identification of functional regulatory sequence alterations, we annotated the ASB-carrying SNVs using data from several databases on phenotype–genotype associations: NHGRI-EBI GWAS catalog[36], ClinVar[37], PheWAS[38], and BROAD fine-mapping catalog of causal autoimmune disease variants[39]. With these data, we counted the number of known associations per SNP, considering SNVs of several classes: low-covered SNVs not tested for ASB (non-candidate sites having the maximal read coverage across experiments not reaching 20); candidate sites that exhibit or not exhibit ASB from the data sets of a single TF; candidate sites from the data sets for two or more TFs that, again, exhibit ASB or do not; and finally, regulator-switching ASBs, where different TFs prefer to bind alternative alleles, e.g., in different cell types. All variants

were segregated into classes in regard to known associations: no known associations, with a single association, and with multiple associations.

We have found that the share of ASB variants with genetic associations was consistently higher than expected by chance (Fig. 5a), which apparently makes ASBs good candidates for prospection for causal SNVs. Specifically, the odds ratio between the observed and expected SNP numbers was specifically high for TF-switching ASBs, although only 1.5% of such ASBs were involved in two or more known GWAS associations. For many variants, there are no known associations with "macro-pheno-types," as provided by GWAS studies, but there are data on molecular phenotypes like variations in mRNA levels. In fact, the effect of the so-called eQTLs[40] can be explained by the alteration of TF-binding affinity that is revealed by ASB. Using the same classification of SNVs as above, we tested ASB and non-ASB SNVs for overlaps with GTEx[41] eQTLs and observed the same pattern as for phenotype associations, with the strongest enrichment of ASBs for which different TFs preferably bind alternative alleles (Fig. 5b). The enrichment also grew stronger with the number of genes, mRNA levels of which were associated with the variant. The same effect holds for multi-cell type ASBs (Supplementary Fig. 9).

More than 80% of ASB SNVs with alternative alleles preferably bound by different TFs overlap eQTLs in at least one cell type, whereas 10% of such ASB SNVs overlap eQTLs targeting ten or more genes. A large fraction of genes of medical relevance from the ClinVar catalog[37] was found among protein-coding genes associated with ASB eQTLs (twofold enrichment as compared to random expectation, Fisher's exact test *P* ~10$^{-49}$). Of note, as many as 90% of genes of medical relevance in ClinVar are eQTL targets of ASB SNVs, and this constitutes 30% of all target genes of ASB eQTLs.

It is not trivial to measure the reliability of ASB identification due to difficulty in assembling a highly reliable "ground truth" set of ASBs, that is necessary to compute standard performance measures based on true and false positives/negatives. For instance, only synthetic data were used for benchmarking purposes in the original BaalChIP paper[19]. On the other hand, despite difficulties in the direct evaluation of ASB calling performance, it is possible to estimate implicitly the "regulatory potential" of particular SNPs from functionally related data. We performed a comparison of ASB calls between ADASTRA, BaalChIP, and Shi et al. data comparing ASBs with GTEx eQTLs (Fig. 5c and Supplementary Fig. 9). The level of eQTL support for ADASTRA ASBs turned out to be comparable to that of the BaalChIP ASB set, with Shi et al. data close behind.

We also studied the association of GWAS-tested phenotypes with all candidate SNVs, not necessarily significant ASBs, found in TF-binding regions. To this end, we performed a general enrichment analysis for SNPs found in ChIP-Seq data of particular TFs within linkage disequilibrium blocks (LD-islands identified in[42]) using Fisher's exact test (see "Methods"). Thus we identified TFs for which phenotype-associated SNVs were enriched within TF-binding regions (Supplementary Fig. 9). For a number of TFs such association with phenotypes was reported in other studies. The examples include FOXA1 (involved in prostate development[43] and in our case, found associated with prostate cancer), IKZF1 (for which the protein damaging mutations are associated with leukemia), STAT1 (involved in the development of systemic lupus erythematosus[44]), and others. Practically in all cases one or several of the associated SNVs also acted as ASBs of the respective TF, providing strong candidates for causality.

To illustrate how the functional role of regulatory SNPs can be highlighted with ASB data, we present several case studies. First, there is rs3761376 (G > A) that serves as a Ref-ASB for ESR1, which was already confirmed by electrophoretic mobility shift assay[45]. rs3761376 is located in the TFF1 gene promoter and was shown to reduce TFF1 expression through altered ESR1 binding, suggesting a molecular mechanism of the increased risk of gastric cancer[45].

Next, there is rs17293632 (C > T) that serves as a Ref-ASB for 25 different TFs and was previously reported to affect the chromatin accessibility in the adjacent region[46]. rs17293632 is associated with Crohn's disease. This SNP is located in SMAD3 intron and overlaps an eQTL targeting SMAD3, AAGAB, and PIAS1 genes[41]. Interestingly, a variant of SMAD3 is also associated with Crohn's disease, particularly, with increased risk of repeated surgery and shorter relapse[47]. Among the TFs displaying ASBs, there are JUN/FOS proteins with the ASB-concordant motif annotation. The AP1 pioneer complex of JUN/FOS likely serves as a "driver" for changes both in gene expression and chromatin accessibility, and is likely to cause ASB of all 25 TFs.

Apart from multi-TF ASBs which are linked to local chromatin changes, non-trivial cases can be found among TF-switching ASBs. For example, SNP rs58726213 is associated with psoriasis and is ASB of CREB1 (reference allele preference, concordant with motif) and JUN (alternative allele preference). rs58726213 is located in the STX4 intron or upstream region depending on a transcript variant. STX4 is significantly downregulated in psoriasis[48], and, according to GTEx, rs58726213 serves as an eQTL of STX4 and HSD3B7; the latter is also reported as psoriasis susceptibility locus[49].

Another example is SNP rs11257655 that is associated with type 2 diabetes[50]. rs11257655 is reported to be located in the CDC123 regulatory region and exhibits ASB of FOXA1 (alternative allele preference, concordant with the sequence motif),

ESR1 (reference allele preference), and three other TFs (SPI1, STAT1, and SMC3). According to UniProt[51], FOXA1 is involved in liver and pancreas development, and in glucose homeostasis. At the same time, polymorphisms in the ESR1 gene are associated with type 2 diabetes and with fasting plasma glucose[52,53].

Thus, ASBs highlight the cases where phenotype–genotype associations arise with different mechanisms, either from protein structure variation, or due to altered gene expression caused by nucleotide substitutions in the gene regulatory region.

## Discussion

The functional annotation of noncoding variants remains a challenge in modern human genetics. Phenotype-associated SNPs found in GWAS are usually located in extensive linkage disequilibrium blocks, and reliable selection of causal variants cannot be done purely by statistical means. Additional data for the identification of causal variants come from functional genomics. In particular, an important class of causal variants consists of regulatory SNVs affecting gene transcription. For those variants, there are various approaches, e.g., parallel reporter assays, to obtain high-throughput data on molecular events caused by particular nucleotide substitution. Another common strategy is to check if a variant of interest falls into a known gene regulatory region detected by chromatin immunoprecipitation or chromatin accessibility assay followed by deep sequencing. By assessing the allele specificity, it is possible to further profit from these data through direct estimation of the effect that a particular allele has on the binding of relevant regulatory proteins or chromatin accessibility.

In this meta-study, for each SNV, we integrated the data by considering a TF bound to SNV in different cell types or a cell type and different TFs bound to the same SNV. Surprisingly, ASB identification through data aggregation had better sensitivity than standard ChIP-Seq peak calling at the level of individual data sets. Particularly, in GTRD, the ChIP-Seq peak calls were gathered from four different tools (MACS, SISSRs, GEM, and PICS), but only 85–90% of significant ASBs were detected within peak calls (199,819 of 233,290 and 324,890 of 351,965 for TF-centric and cell type-centric aggregation), suggesting that up to 15% of ASBs could be lost if the ASB calling was restricted to the peak calls only.

Each particular ASB can either be a "driver" directly altering TF-binding affinity, or a "passenger" with differential binding resulting from differential chromatin accessibility (in turn, caused by some neighboring SNVs), or a protein–protein interaction with the causal TF. In terms of machine learning, we expected the TF ASBs to provide an easier prediction target since they could be mostly determined by the sequence motif of the respective TF. However, as found, the percentage of "passenger" ASBs is rather large (e.g., 24,662 out of 27,233 CTCF ASBs lack significant CTCF motif hits), and the TF-specific models showed a limited ASB prediction quality. Further surprise came from cell type-specific models which displayed a notably higher performance. We interpret these data as follows: the cell type ASBs are easier to predict by learning a small set of cell type-specific master regulators, while passenger TF-level ASBs are very diverse, as coming from data aggregation of many different cell types with varying cell type-specific features such as key TFs.

ASB events should be distinguished from other sources of allelic imbalance such as aneuploidy and local CNVs, which can imitate ASB by varying the allelic dosage. Commonly used cell types are often aneuploid: K562 and MCF7 cells are triploid on average, and 59 of 121 cell types overlapping between ADASTRA and COSMIC also have median copy number above 2. The ADASTRA pipeline, to our knowledge, includes the first control-free approach to

reconstruct the genomic map of BAD directly from SNP calls and to use this map as a baseline for detecting genuine allelic imbalance. Despite a multitude of available software for ASB calling, there has been no approach suitable for the uniform analysis of diverse existing data. Thus, when developing ADASTRA, the intention was to be able to process and include most of the data including non-replicated experiments, data sets lacking genomic input controls, or with the controls sequenced at low coverage, at the expense of general sensitivity achieved at particular data sets. Further on, such a pipeline might be applicable to other sequencing data that allow allele specificity, e.g., analyses of allele-specific expression or chromatin accessibility. With matched cell types, BAD-corrected data on allele-specific chromatin accessibility will also allow for better classification of driver and passenger ASBs and better application of machine learning techniques.

Our collection of ASB events per se is also useful for other research areas involving TF–DNA interactions. First, ASBs provide unique in vivo data on differential TF binding and can be used for testing the predictive power of computational models for precise recognition of TF-binding sites[33]. Second, the TF binding not only affects transcript abundance, but also affects RNA splicing, localization, and stability[54,55]. Thus, ASBs may affect other levels of gene expression, particularly, the mRNA post-transcriptional modification: out of 65 RNAe-QTLs reported in[56], 4 are listed as ASBs in ADASTRA.

Last but not least, ADASTRA reports hundreds of TF-switching ASBs, where alternative alleles are preferably bound by different TFs. This possibility has been discussed previously[57] but, to our knowledge, we are first to report the genome-wide inventory of such events. Importantly, the respective SNVs exhibit the highest enrichment with phenotype associations. Probably these sites serve varying and allele-dependent molecular circuits. A particularly interesting example is rs28372852 located in the G elongation factor mitochondrial 1 (GFM1) gene promoter. According to ADASTRA, rs28372852 serves as the Alt-ASB of CREB1 and Ref-ASB of MXI1, and in both cases, the allelic imbalance is concordant with the respective binding motifs. Also, according to GTEx[41], GFM1 is the target of rs28372852 eQTL. According to UniProt[48], CREB1 is a transcriptional activator, while MXI1 is a transcriptional repressor, suggesting that ASB can directly switch the gene expression activity. At the same time, UniProt reports four amino acid substitutions in GFM1 that are associated with combined oxidative phosphorylation deficiency. Interestingly, according to ClinVar[37], this SNP is benign in regard to combined oxidative phosphorylation deficiency; and in this case we speculate that ASB data might facilitate reevaluating the variants' functional roles and pathogenic potential. We believe that further analysis of TF-switching ASBs in the scope of metabolic and regulatory pathway alterations will provide valuable insights into molecular mechanisms underlying particular normal and pathologic traits.

## Methods

**Variant calling from GTRD alignments**. We used 7669 premade short read alignments against hg38 genome assembly produced with bowtie2[58] and stored in the GTRD[17] database. PICARD was used for deduplication, followed by GATK base quality recalibration. Next, the variants were called with GATK Haplotype-Caller, with dbSNP[26] (common variant set of the build 151) for annotation. The resulting variant calls were filtered to meet the following requirements: (1) an SNV must be biallelic and heterozygous (GATK annotation GT = 0/1); (2) an SNV must have read coverage ≥ 5 at both the reference and alternative alleles); (3) an SNV must be listed as an SNP in the dbSNP 151 common set. Of note, we considered all eligible SNVs as candidate ASB, not necessarily located within ChIP-Seq peak calls.

We restricted ourselves with variants from the dbSNP common subset due to the following reasons: (1) allelic read counts at de novo mutations reflect the composition of the cell population (i.e., the fraction of cells carrying the mutation) rather than the local copy-number ratio or ASB; (2) de novo point mutations within particular copies of duplicated segments (considering, e.g., chromosome

duplications) will exhibit allelic imbalance (e.g., in a tetraploid region with 2:2 ratio of allelic reads at SNPs, de novo mutations will likely exhibit the ratio of 1:3) and may lead to false-positive ASB calls.

**Accounting for BAD**. The observed distribution of ChIP-Seq allelic read counts on heterozygous SNVs significantly depends on aneuploidy and the CNV profile of the cells (Fig. 6a, b). The modes of distribution correspond to the most represented copy number, e.g., the distribution is bimodal for mostly triploid K562 cells, Fig. 6b. However, the mixture of two Binomial distributions poorly approximates the data, showing a significant overdispersion. To systematically reduce the overdispersion from local CNVs and aneuploidy, we reconstructed the genome-wide BAD maps from read counts at the heterozygous variants (see below). The distributions of the allelic read counts at SNVs segregated by BAD show a notably reduced overdispersion (Fig. 6c, d).

**BAD calling with Bayesian changepoint identification**. To construct genome-wide BAD maps from filtered heterozygous SNV calls, we developed a novel algorithm, the BAD caller by Bayesian changepoint identification (BABACHI).

At the first stage, BABACHI divides the chromosomes into smaller sub-chromosome regions by detecting centromeric regions, long deletions, loss of heterozygosity regions, and other regions depleted of SNVs. At this stage, only the distances between neighboring SNVs are taken into account and long gaps are marked. The sub-chromosome regions with <3 SNVs or chromosomes with <100 SNVs are removed. Next, BABACHI finds a set of changepoints in each sub-chromosome region that further divide it into smaller segments of stable BAD. The optimal changepoints are chosen to maximize the marginal likelihood to observe the experimental distribution of allelic read counts at the SNVs, given a region-specific (yet unknown) BAD persist in each region enclosed between neighboring changepoints. Finally, a particular BAD is assigned to each segment according to the maximum posterior.

The likelihood is calculated for the statistic $x = \min(C_{Ref}, C_{Alt})$, assuming $C_{Ref}$ to be distributed according to the truncated Binomial distribution ~TruncatedBinom $(n, p)$ given that $C_{Ref} + C_{Alt} = n$, the number of reads overlapping the variant; the number of successes $k$ is limited to $5 \leq k \leq n-5$ (the read coverage filter), and $p$ is either $1/(BAD + 1)$ or $BAD/(BAD + 1)$, matching one of the expected allelic read frequencies.

BAD of each segment is selected from the discrete set {1, 4/3, 3/2, 2, 5/2, 3, 4, 5, 6}, considering that the total copy number of a particular genomic region rarely exceeds 7. The prior distribution of BAD is assumed to be a discrete uniform, with the support being the same discrete set as above (non-informative prior). Details and mathematical substantiation of the algorithm are provided in the Supplementary Methods.

**Practical BAD calling with the ADASTRA pipeline**. To provide better genome coverage and robust BAD estimates, we merged the sets of variant calls from ChIP-Seq data sets produced in the same laboratory for the same cell type and in the same series (i.e., sharing either ENCODE biosample or GEO GSE ID). Different SNVs at the same genome position (either originating from different data sets or with different alternative alleles) were considered as independent observations. For each data set, chromosomes with < 100 SNVs were excluded from BAD calling and further analysis.

To assess the reliability of the BAD maps, for each BAD, we separately estimated ROC and PRC. Here we considered the BAD maps as binary classifiers of SNVs according to BAD, with COSMIC CNV data as the ground truth. To plot a curve for BAD = x, the following prediction score was used:
$$S = L(BAD = x) - \max_{y \neq x} L(BAD = y),$$ where $L$ denotes the log-likelihood of the segment containing the SNV to have the specified BAD (Fig. 2c, d).

**Construction of an independent BAD map for MCF7 cells**. The paired-end reads of MCF7 deep genome sequencing (SRA accession SRR8652105) were aligned to hg38 genome assembly using bowtie2 with default settings. Overall, 28,278,026 (2.5%) of a total of 1,136,666,560 paired reads were marked as duplicates, 112,323,925 (9.9%) were filtered by GATK filter by mapping quality ≥ 10, leaving 996,064,609 reads for SNP calling. A total of 3,969,250 SNPs was reported by GATK HaplotypeCaller, among which 1,427,492 SNPs were annotated as heterozygous, passed the basic ADASTRA filter (≥5 reads on each allele), and were used to produce the independent reference MCF7 BAD map with BABACHI[69].

**ASB calling with the Negative Binomial mixture model**. To account for mapping bias, we fitted separate Negative Binomial mixture models for the scoring of Ref- and Alt-ASBs. For each BAD and each fixed read count at Ref- and Alt- alleles, we obtained separate fits using SNVs from all available data sets.

For every fixed read count value at a particular allele, we approximated the distribution of read counts mapped to the other allele as a mixture of two Negative Binomial distributions. The model estimates the number of successes $x$ (the number of reads mapped to the selected allele) given the number of failures $r$ (the number of reads mapped to the second allele) in the series of Bernoulli trials with probability of success $p$ (for the first distribution in the mixture) or $1 - p$ (for the second distribution in the mixture). The following holds for scoring Ref-ASBs at

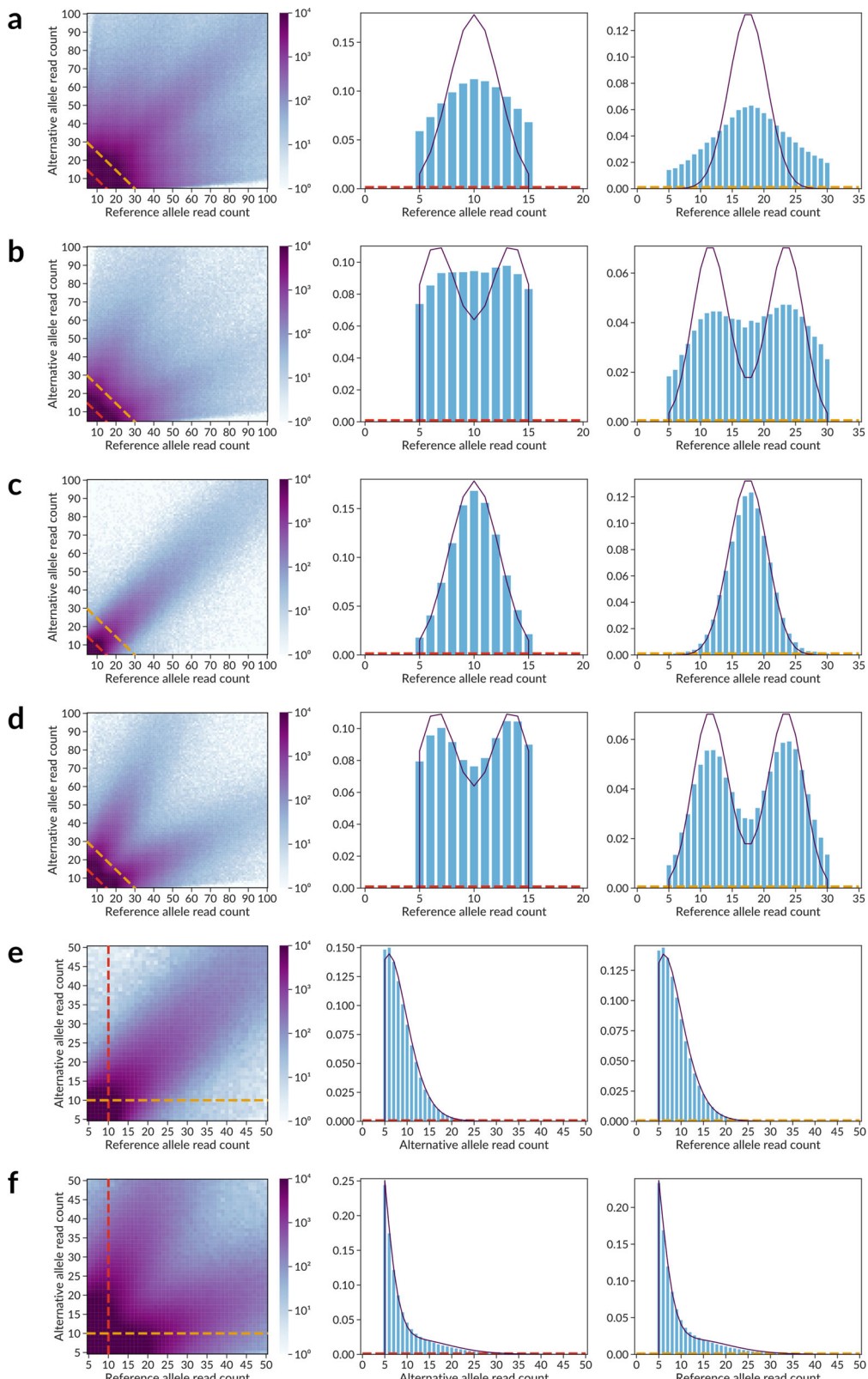

fixed Alt-allele read counts:

$$C_{\text{Ref}}|\text{fixed } C_{\text{Alt}} \sim (1-w) \times \text{NegativeBinomial}(r, p) + w \times \text{NegativeBinomial}(r, 1-p)$$

$$P(C_{\text{Ref}} = x | \text{fixed } C_{\text{Alt}} = m, C_{\text{Ref}} \geq 5)$$

$$= \binom{x+r-1}{x}\left((1-w) \times (1-p)^r \times p^x + w \times (1-p)^x \times p^r\right)/A \quad (1)$$

$$A = 1 - P\left(C_{\text{Ref}} < 5 | \text{fixed } C_{\text{Alt}} = m\right)$$

where $p$ and $1-p$ were fixed to reflect the expected frequencies of allelic reads, namely, $1/(\text{BAD}+1)$ and $\text{BAD}/(\text{BAD}+1)$. The parameters $r$ (number of failures) and $w$ (weights of distributions in the mixture) were fitted with L-BFGS-B algorithm from *scipy.optimize*[59] package to maximize the model likelihood iteratively with boundaries $r > 0$ and $0 \leq w \leq 1$, assigning initial values of $r = m$ (number of reads on the fixed allele) and $w = 0.5$, respectively. $A$ is the normalization coefficient (necessary due to truncation) corresponding to allelic reads cutoff of 5. The goodness of fit was assessed by root mean square error of

**Fig. 6 Distribution of read counts at SNVs significantly depends on background allelic dosage.** Each panel contains three plots: (1, left) a heatmap of allelic read counts colored by $\log_{10}$[number of SNVs that have the specified number of ChIP-Seq reads] supporting the reference ($X$-axis) and alternative ($Y$-axis) alleles; (2, middle, 3, right) barplots of observed read counts at one of the alleles and the approximating distribution plot. Two barplots correspond to the two slices of the heatmap data, either by fixing the sum of reads at two alleles (**a**–**d**, diagonal slices along the dashed lines in the bottom left corner, approximated by the binomial mixture) or by fixing the read counts at one of the alleles (**e**, **f**, vertical and horizontal slices, approximated by the negative binomial mixture). **a** Complete set of ADASTRA candidate ASB SNVs, no separation by BAD, the observed distribution can be interpreted as overdispersed binomial. **b** K562 candidate SNVs, the distribution is similar to an overdispersed mixture of binomial distributions with $p = \frac{1}{3}$ and $p = \frac{2}{3}$ as K562 are mostly triploid. **c** SNVs in diploid regions according to BAD = 1, binomial distribution with $p = \frac{1}{2}$. **d** SNVs in BAD-separated triploid regions (BAD = 2), binomial mixture with $p = \frac{1}{3}$ and $p = \frac{2}{3}$. **e** BAD-separated diploids (BAD = 1), negative binomial distribution with $p = \frac{1}{1}$ (fit). **f** BAD-separated triploids (BAD = 2), negative binomial mixture with $p = \frac{1}{3}$ and $p = \frac{2}{3}$ (fit). In all the cases the distributions are truncated, corresponding to the allelic read counts cutoff of 5. SNV single-nucleotide variant, ASB allele-specific binding, BAD background allelic dosage.

approximation (RMSEA[60], Supplementary Fig. 11). Low-quality fits with RMSEA > 0.05 were discarded, fixing the parameters at $r = m$ and $w = 1$, thereby penalizing the statistical significance of ASB at such SNVs, as fitted $r$ is systematically lower than $m$ (Supplementary Fig. 12). Of note, the values of $r$ for distribution of reference allele read counts (with fixed alt-allele read counts) were systematically higher than those for alternative allele read counts (with fixed Ref-allele read counts), thus balancing the reference mapping bias. The obtained fitted models were used for statistical evaluation of ASB for alternative and reference alleles independently, with one-tailed tests. Examples of fits for BAD = 1 and 2 are shown in Fig. 6e, f, with RMSEA < 0.02 for the fixed Ref/Alt read counts of 10.

**Aggregation of ASB $P$ values from individual data sets.** For each ChIP-Seq read alignment (except control data), we performed the ASB calling. Next, the SNVs were grouped by a particular TF (across cell types) or by a particular cell type (across TFs). A group of SNVs with the same position and alternative alleles was considered as an ASB candidate if at least one of the SNVs passed a total coverage threshold ≥ 20. Next, for each ASB candidate, we performed *logit* aggregation of individual ASB $P$ values[27], independently for Ref-ASB and Alt-ASB. Individual $P$ values of 1 were excluded from aggregation, and if none were left, the aggregated $P$ value for an SNV was set to 1.

Logit aggregation is the method of a choice, as it has two advantages. First, compared to Fisher's method, it cancels out symmetrical $P$ values like 0.01 and 0.99 to 0.5. Second, the pattern of evidence is not known in advance, significant ASB $P$ values can arise both from a small number of strongly imbalanced SNVs in deeply sequenced data sets and from a large number of weakly imbalanced SNVs in data sets with low or medium coverage. Compared to the similar Stauffer's method, the logit aggregation is less sensitive to the extreme $P$ values and can be considered a robust choice[61]. The resulting aggregated $P$ values were FDR corrected (Benjamini–Hochberg adjustment) for multiple tested SNVs separately for each TF and each cell type. SNVs passing 0.05 FDR for either Ref or Alt-allele were considered ASB.

**ASB effect size estimation.** We define the ES separately for reference allele ASB ($\mathrm{ES_{Ref}}$) and alternative allele ASB ($\mathrm{ES_{Alt}}$) as the log ratio of the observed number of reads to the expected number. To account for BAD and mapping bias, we use fitted Negative Binomial mixture at the fixed allele read counts:

$$\mathrm{ES_{Ref}} = \log_2\left(C_{Ref}/E\left(C_{Ref}|C_{Alt}\right)\right),$$
$$\mathrm{ES_{Alt}} = \log_2\left(C_{Alt}/E\left(C_{Alt}|C_{Ref}\right)\right) \quad (2)$$

In the basic case of BAD = 1, the ES can be approximated as the log ratio of read counts, taking into account that the expectation bias due to the truncation is relatively small and $r$ is close to the read count on the fixed allele: $\mathrm{ES_{Ref}} \approx \log_2(C_{Ref}/C_{Alt})$.

In the case of BAD > 1, the same assumptions lead to the following estimation of the ES:

$$\log_2(C_{Ref} \times \mathrm{BAD}/C_{Alt}) \lesssim \mathrm{ES_{Ref}} \lesssim \log_2(C_{Ref}/(\mathrm{BAD} \times C_{Alt})) \quad (3)$$

This holds due to the fact that for fixed BAD, $C_{Ref}$ expectation is either $C_{Alt} \times \mathrm{BAD}$ or $C_{Alt}/\mathrm{BAD}$, depending on a haplotype. Therefore, the expectation of $C_{Ref}$ according to the Negative Binomial mixture model is approximately $w \times C_{Alt} \times \mathrm{BAD} + (1 - w) \times C_{Alt}/\mathrm{BAD}$.

The final ASB ES is estimated for SNVs with aggregated significance either across TFs or across cell types. The ES value is calculated as a weighted average of ES of individual SNVs in aggregation, with weights assigned as negative logarithms of individual $P$ values. ES is not assigned in the case if all individual $P$ values are equal to 1.

**SNV and ASB annotation**

*Genomic annotation.* To annotate SNVs according to their genomic location (Fig. 3c), we started with mapping SNVs to FANTOM5 enhancers and promoters[62]. The remaining SNVs were annotated with ChIPseeker[63] with a hierarchical assignment of the following categories: promoter (≤1 kb), promoter

(1–2 kb), promoter (2–3 kb), 5′UTR, 3′UTR, Exon, Intron, Downstream, Intergenic. For clarity, promoter (≤1 kb) and 5′UTR categories were both tagged as "promoter"; promoter (1–2 kb) and promoter (2–3 kb) were both tagged as "upstream."

*Sequence motif analysis of ASBs.* For TF ASBs, we annotated the corresponding SNVs with sequence motif hits of the respective TFs. To this end, we used models from HOCOMOCO v11 core collection[64] and SPRY-SARUS[65] for motif finding. The top-scoring motif hit was taken considering both Ref and Alt alleles, and, at this fixed position, the "motif FC" was calculated as the $\log_2$-ratio of motif $P$ values at the reference and alternative variants so that the positive FC corresponded to the preference of the alternative allele.

To analyze the ASB motif concordance (Fig. 4), we considered the ASB SNVs (min($\mathrm{FDR_{Ref}}$, $\mathrm{FDR_{Alt}}$) ≤ 0.05) that overlapped the predicted TF-binding site: (min (motif $P$ value$_{Ref}$, motif $P$ value$_{Alt}$) ≤ 0.0005), and had |FC| ≥ 2. We defined the motif concordance/discordance as a match/mismatch of the signs of FC and $\Delta\mathrm{FDR} = \log_{10}(\mathrm{FDR_{Alt}}) - \log_{10}(\mathrm{FDR_{Ref}})$.

*Annotation of ASBs with phenotype associations.* To assess enrichment of ASBs within phenotype-associated SNPs, we used the data from four different SNP-phenotype associations databases, namely: (1) NHGRI-EBI GWAS catalog[36], release 8/27/2019 with EFO mappings[66] used to group phenotypes by their parent terms for Supplementary Fig. 9; (2) ClinVar catalog[37], release 9/05/2019 (entries with "likely pathogenic," "pathogenic," or "risk factor" clinical significance); (3) PheWAS catalog[38]; (4) BROAD fine-mapping catalog of causal autoimmune disease variants[39]. All entries were systematized in the form of triples <dbSNP ID, phenotype, database>. Next, the entries were annotated with the TF- or cell type-ASB data.

To evaluate TF-phenotype associations in detail, we used NHGRI-EBI GWAS catalog and the following pipeline:

(1) We filtered out TFs with less than two candidate ASBs, and phenotypes associated with less than two SNPs, resulting in 765 TFs and 2688 phenotypes suitable for the analysis. For each TF, we considered all SNPs with candidate ASBs passing the coverage thresholds.

(2) For each pair of a TF and a phenotype, we calculated the odds ratio and the $P$ value of the one-tailed Fisher's exact test on SNPs with candidate ASBs considering two binary features: whether the SNP is associated with the phenotype, and whether the SNP is included in ASB candidates of the particular TF. The superset of SNPs was collected independently for each TF by gathering SNPs with candidate ASBs for all TFs but only from LD blocks[42] containing either TF-specific SNPs or phenotype-associated SNPs. The $P$ values were then FDR corrected for multiple tested TFs separately for each phenotype.

*Analysis of eQTLs and eQTL target genes.* To analyze an overlap between ASBs and eQTLs, we used significant <variant, gene> pairs from GTEx[41] (release V8).

To evaluate ASB-driven eQTL target genes' associations with medical phenotypes, a one-tailed Fisher's exact test was performed on the enrichment of protein-coding genes of medical relevance (6026 genes found linked with entries with "pathogenic," "likely pathogenic," or "risk factor" clinical significance in ClinVar catalog[37]) among eQTL target genes of ASB SNPs (16,865 protein-coding genes according to GTEx), considering all human protein-coding genes from GENCODE[67] (v35, 19,929 gene symbols) as the background set.

**ASB prediction with machine learning.** In our work, we used a standard software implementation of the random forest model from the scikit-learn package. The number of estimators was set to 500 and the other parameters were defaults. Three feature types were used (Supplementary Table 4): allele-specific chromatin DNase accessibility, synthetic data from neurons from the last layer of the DeepSEA[11], and HOCOMOCO motif predictions obtained with SPRY-SARUS[65]. As a global set of SNVs, we used 231,355 dbSNP IDs overlapping between ADASTRA and Maurano

et al.[16] data, which provided allele-specific DNase accessibility. For the general model, we used SNVs with ASBs for any of TFs or in any of cell types as members of the positive class, and the remaining set of candidate SNVs as members of the negative class. For TF- and cell type-specific assessment, we defined ASB and non-ASB SNVs for a particular TF or in a particular cell type as the positive and negative class, respectively.

**Reporting summary**. Further information on research design is available in the Nature Research Reporting Summary linked to this article.

## Data availability
The complete data on ASBs across TFs and cell types described in this study are available in the release 1.6.10-Soos of the ADASTRA database (http://adastra.autosome.ru/) and provided online: http://adastra.autosome.ru/soos/, the generated BAD maps and the list of ChIP-Seq data sets are available at http://adastra.autosome.ru/soos/downloads. The reprocessed ChIP-Seq peaks and metadata are available in the GTRD database: http://gtrd.biouml.org.

## Code availability
The ADASTRA pipeline is available at GitHub: https://github.com/autosome-ru/ADASTRA-pipeline[68]. BABACHI segmentation software is available at GitHub: https://github.com/autosome-ru/BABACHI[69]. The code for machine learning analysis is available at GitHub: https://github.com/autosome-ru/ASB-ML[70]. The SPRY-SARUS motif scanner is available at GitHub: https://github.com/autosome-ru/sarus[65].

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

## Acknowledgements

We thank the organizers and members of the GRECO consortium for the series of workshops (held under European Union COST Action CA15205—GREEKC, coordinator Martin Kuiper) which provided a fruitful networking and discussion platform for ideas of this study. We personally thank Denis Litvinov for help in GTRD metadata processing and Evgenia Serebrova for help in paper preparation. This study was supported by RFBR grant 18-34-20024 to I.V.K. (basic ADASTRA pipeline), RSF grant 20-74-10075 to I.V.K. (machine learning and additional analysis), RSF grant 19-14-00295 to F.K. (GTRD data extraction).

## Author contributions

S.A. and A.B. developed the computational framework and database; S.A., A.B., and I.E.V. developed the website; D.B., E.B., I.E.V., A.V.F., and M.V.F. performed the functional annotation and motif annotation of ASBs; D.B. and D.D.P. performed the machine learning analysis; I.Y., S.K.K., and F.K. established the GTRD alignments processing; V.J.M. and I.V.K. designed and supervised the study. All authors participated in the paper preparation.

## Competing interests

The authors declare no competing interests.
