## [Peer Review File · Nature Communications]

REVIEWER COMMENTS

Reviewer #1 (Remarks to the Author):

In this study, Abramov et al. present an approach that allows calling the allele-specific transcription factor (TF) binding events at single-nucleotide variants (SNVs) in ChIP-Seq data accounting for the joint contribution of aneuploidy, local copy number variation (CNVs) (estimated directly from the variant call) and read mapping bias. Using the suggested framework, the authors analyzed 7669 ChIP-Seq experiments from the GTRD database, identified allele-specific binding events (ASBs), and assembled the ADASTRA1 database. This resource contains more than 500K entries at nearly 270K SNVs for 674 human TFs (including epigenetic factors) and 337 cell types.

Already existing computational and in vitro approaches allow assessing the effects of SNVs on TF binding or gene expression alterations. However, these methods tend to have difficulty in uncovering the involved TFs (e.g. massively parallel reporter assays) or in functionally annotating SNV effects at a genome-wide scale in a cell-type-specific manner. To address these issues, the authors suggest using multiple ChIP-Seq data to study the functional effects of SNVs at heterozygous loci, where TFs differentially bind to alternative SNV alleles in homologous chromosomes. Indeed, ChIP-Seq data has proven to be a valuable resource for studying genetic variability, especially when aggregating ChIP-Seq data from multiple experiments.

Some of the already developed approaches for ASB detection account for various types of biases, such as the allelic mapping balance, confounding effects arising from CNVs, etc. To account for biases in estimations of allelic frequencies introduced by CNVs and aneuploidy, the authors introduce the Background Allelic Dosage (BAD) and Bayesian Change-point Identification (BABACHI) algorithms to construct genome-wide BAD maps, used as a baseline for detecting allelic imbalance. One of the merits of this work is that the authors use the suggested approach not only to find ASB events at multiple SNVs but also use these results to assemble the ADASTRA database. The latter has a user-friendly interface and appears to be a practical resource containing functional annotations of non-coding variants across multiple cell types. This in turn has the potential to bring us one step closer to understanding the effects of variants in gene regulatory regions and their contribution to complex traits.

However, the paper would benefit substantially from clarifying parts of the underlying methodology and presenting the analysis, as well as the results, in a more biologically relevant context, especially in the machine learning part. It was nice to see that we have significant power of predicting ASBs from presumably correlated measurements, but without yielding mechanistic insights, these findings add little to the field as a whole.

Comments:

Conceptual:

- Does the suggested BAD approach make a striking difference compared to traditional ASB methods, especially BaalChIP2? In an age where the emergence of new software tools is rampant, proper benchmarking of a new approach against already existing ones is absolutely key to guide downstream users toward the best possible approach in function of specific data types. As it stands, the study does not include any benchmarking, which should be addressed.
- The statement about MCF7's genome instability is confusing in the context of rationalizing discrepancies with the BAD results. While there may certainly be genetic differences between lab lines, these generally tend to be rather limited. Moreover, the corresponding ChIP-Seq was performed on millions of cells which thus must represent the "true" meta genotype and not that of individual subclones. These discrepancies need therefore to be investigated in more detail as it undermines at this point the accuracy of the BAD approach. The next statement "Similar results were obtained additional comparison of BAD maps against independent microarray-based CNV estimates" is also not clear -- is it similar to results being different or the overall prediction being similar to the reported CNVs?

- To allow more insights into the performance of the overall workflow, it would be helpful to provide a side by side comparison of ASBs between called events in this study and those published earlier for some TFs in well-characterized cell lines, such as K562 or LCL.

- The authors infer BAD under the assumption that the majority of SNVs in a region are independent of each other, saying that an ASB event in one SNV does not imply ASB somewhere nearby. However, given the increasing body of literature that attributes an important role to TFs and other proteins in steering local genome organization by aggregation, this assumption does not necessarily hold. Where TF binding shows such coordinated effects, BADs might thus be wrongly inferred, resulting in a substantial portion of false negatives. Indeed, the authors list that their ASB calling includes “only” 38 – 63% of ASBs reported elsewhere. However, it is not obvious from the text why ADASTRAs is missing up to 60% of ASBs from existing sources, especially since one can assume that most ASBs that are inferred from smaller datasets should be included in the total set of ASB calls from a larger resource. One major cause for this high number of false negatives could be the reason detailed above, but there may be other ones, all of which should be carefully investigated, as they may undermine the conceptual robustness of the presented workflow.

- Do the authors suggest that the ASB distance distribution is bimodal with the shorter distance falling within the typical ChIP-seq peak width because more reads fall in that region and thus improve statistical power? The reason for observing such a bimodal distribution is not obvious to the reader and should be clarified.

- Why did the authors use the ratio of log p-values for their motif analysis instead of using the relative affinities as obtained directly from the motif scores?

- The general concordance of motif presence/absence and ASB is encouraging and ADASTRAs should thus provide a valuable resource for further, more in-depth studies. In this regard, it would be interesting to see if there are any examples of TFs that have sufficient motif variations overlapping ASBs that a semi-quantitative comparison of ASB against motif score would be possible?

- From the main text, it remains unclear what exactly the two binary classification problems for machine learning are. The authors should clarify this section. Besides, a biological motivation why specific features are chosen would greatly help the reader to better understand the section. For instance, why are p-values instead of direct TF motif scores used (see also above)? In particular, a summary of the importance of the individual features would justify why this section is in the paper. ATAC-seq differential accessibility is expected to correlate with TF ASB, given that for many TFs the majority of peaks fall within accessible regions. Thus, differentiating between TFs that are capable/not capable of binding to inaccessible regions could provide useful information on the importance of TF binding affinity to their cognate sites.

Technical:

- How many ASBs are expected to be missed because of the filtering procedures?

- How many cell lines are in fact majorly affected by aneuploidy?

- How many new ASB events were identified due to the suggested BAD approach? Does one find many more ASB events compared to the already existing approaches?

- In the section “Motif annotation is concordant with ASB calls” the authors state that for “~10-20% of SNVs more reads are attracted to the weaker motif hit”. It would be nice to get some additional insights here and identify TFs that share the same ABSs, for instance. Additionally, it would be interesting to see how many TFs are there with a clear motif delta but with no ASBs.

- The description of the staveplot remains unclear and should be clarified. Therefore, the statement about less conserved positions allowing for more options is arguable, especially considering sites neighboring to position 6.

References

1. Allelic Dosage-corrected Allele-Specific human Transcription factor binding sites, <http://adastra.autosome.ru>
2. de Santiago, Ines, et al. "BaalChIP: Bayesian analysis of allele-specific transcription factor binding in cancer genomes." *Genome biology* 18.1 (2017): 39.

Reviewer #2 (Remarks to the Author):

This is a very interesting paper describing a novel approach to call the allele-specific transcription factor binding events at SNVs in ChIP-Seq data. The authors have developed new algorithms and software, all of which is publicly available. They have also conducted a very thorough analyses of more than 7000 ChIP-Seq experiments and assembled all of the results into the ADASTRA database. They also compared their annotations with variants in multiple public databases regarding disease/phenotype associations, such as the GWAS catalog and ClinVar. These analyses demonstrate the potential clinical and phenotypic relationships with these transcription factor binding variants. The resources, information, and software presented in this manuscript will be incredibly useful for the field of human genetics.

While the work is extremely thorough and the manuscript very well written, I have a few minor comments that warrant further explanation or detail:

- On line 126, the authors say that they filtered the variant calls by "low-covered variants". I could not find how they define low-covered variants. I think it is important for the reader to know what precise definition was used here.
- On line 139, the authors indicate that "3000 ASBs for cell types passing an adjusted p-value of 0.05". How was this adjustment made? Was this a strict Bonferroni adjustment? Or some other type of multiple testing adjustment?
- On line 576, there is an equation where there are only rectangles in both the numerator and denominator.

Response to reviewers' comments - Reviewer #1

... However, the paper would benefit substantially from clarifying parts of the underlying methodology and presenting the analysis, as well as the results, in a more biologically relevant context, especially in the machine learning part. It was nice to see that we have significant power of predicting ASBs from presumably correlated measurements, but without yielding mechanistic insights, these findings add little to the field as a whole.

We did our best to improve the paper by following all the questions and suggestions, including comparison and benchmarking against BaalChIP data set (ASB calling section) and an overview of the feature importance estimates (machine learning application). Please find more details in a point-by-point response below. Line numbers shown in bold below correspond to those of the revised manuscript.

Does the suggested BAD approach make a striking difference compared to traditional ASB methods, especially BaalChIP2? In an age where the emergence of new software tools is rampant, proper benchmarking of a new approach against already existing ones is absolutely key to guide downstream users toward the best possible approach in function of specific data types. As it stands, the study does not include any benchmarking, which should be addressed.

First, despite a multitude of available software and pipelines for ASB calling, there have been no approach suitable for the uniform analysis of diverse existing data, as we explained in the Introduction (see **line 79-106** in the revised version of the manuscript). Thus, when developing ADAstra, the intention was to be able to process and include most of the data (e.g. including non-replicated experiments, and ChIP-Seq data sets lacking input controls, or with the input control sequenced at low coverage), at the expense of general sensitivity achieved at particular data sets. We have added a necessary clarification in the Discussion (see **line 523-535**).

Second, we fully agree that a comparative assessment of the resulting ASB calls is very instructive. However, we would avoid naming such assessment as 'benchmarking', since by definition the benchmarking requires a well-defined set of performance measures, e.g. proper definition of true and false positives/negatives, obtained at highly reliable 'gold standard' data sets. There are no such data in the field of ASB calling as it is not trivial to perform a large-scale assessment of allele-specific protein binding without explicit usage of ChIP-Seq data. For instance, only synthetic data was used for testing in the original BaalChIP paper.

On the other hand, despite difficulties in the direct evaluation of ASB calling performance, it is possible to estimate implicitly the 'regulatory potential' of particular SNPs from functionally related data. GTEx eQTLs provide a large-scale data set, that we utilized for comparison of ASB calls between ADAstra, BaalChIP, and Shi et al. data, see **line 434-443** and **Fig. 5 panel C** (as well as **Supplementary Fig. S9E**). The general level of eQTL support for ADAstra ASBs turned out to be at a similar level as for the BaalChIP data set, with Shi et al. data showing a bit lower performance.

The statement about MCF7's genome instability is confusing in the context of rationalizing discrepancies with the BAD results. While there may certainly be genetic differences between lab lines, these generally tend to be rather limited. Moreover, the corresponding ChIP-Seq was performed on millions of cells which thus must represent the "true" meta genotype and not that of individual subclones. These discrepancies need therefore to be investigated in more detail as it undermines at this point the accuracy of the BAD approach.

We are interested in genome structural variations primarily because it is the most likely reason for unbalanced allelic dosage. For most of the cell types, our BAD estimates agree with COSMIC copy number data, which confirms the biological relevance of our method. Yet, indeed, this agreement is much worse for MCF7. The reviewer is absolutely correct in his critical remark that the accuracy of the BAD approach needs some special attention in the case of MCF7 cells.

In response to the critique of the reviewer, we have tested if the disagreement between COSMIC copy number map of MCF7 and our BAD calls is related to misidentification of allelic dosage within the ADAstra pipeline. To this end, we extracted independent genomic sequencing data for MCF7 (SRA accession SRR8652105). With these deep sequencing data, we performed read mapping and SNP calling following the ADAstra pipeline and applied BABACHI to construct the BAD map from the genome-wide SNP calls not dependent on ChIP enrichment. The ChIP-independent BAD map agreed reasonably well with MCF7 BAD data obtained from ChIP enriched data sets. Thus, the ADAstra pipeline seems to correctly identify the allelic dosage from ChIP-Seq data for MCF7 cells.

It is noteworthy, that this ChIP-independent map of allelic dosage for MCF7 still rather poorly agrees with the COSMIC CNV markup. We have no ultimate explanation for this observation but would like to remark that MCF7 was found among the most unstable cell types in Ref. 29, with clear sublines found even in the frozen cell bank samples [doi:10.1038/srep28994], and displays rapid modification of copy number alteration profiles [doi:10.1186/1471-2407-3-13]. Yet, due to subclone variability or by some other reason more related to the technicalities of our pipeline (e.g. simple read mapping procedure), it turns out that the COSMIC copy number markup makes a poor proxy for the background allelic dosage for MCF7. Probably, there might be other cell types of the kind, which additionally justifies BAD calling directly from ChIP-seq data, rather than using a fixed reference like a COSMIC copy number map. We have added a new panel to **Supplementary Fig. S2**, and discuss this issue in the manuscript (**line 195-212**).

The next statement "Similar results were obtained additional comparison of BAD maps against independent microarray-based CNV estimates" is also not clear -- is it similar to results being different or the overall prediction being similar to the reported CNVs?

We reconsidered this statement in the revised version of the manuscript. In fact, **Supplementary Fig. S3A** clearly shows that the general agreement between BAD maps

and COSMIC is of the same magnitude as between aCGH of Varma et al. and COSMIC. Of note, in the latter comparison, the performance of aCGH estimate for K562 and MCF7 is reversed relative to our BAD maps, with a stable correlation around 0.5 for MCF7 and poor correlation for K562. We have modified the respective statement in the manuscript accordingly (**line 213-220**).

To allow more insights into the performance of the overall workflow, it would be helpful to provide a side by side comparison of ASBs between called events in this study and those published earlier for some TFs in well-characterized cell lines, such as K562 or LCL.

It is indeed an important question, to which extent ADAstra captures ASBs found in other studies. An ASB could be missed at the SNP calling stage, could fail to pass the coverage filtering threshold, or fail to pass the significance threshold (for FDR-corrected P-value against BAD). To assess the contribution of different stages of our pipeline to ASB calling sensitivity, we took a set of ASB SNPs identified by BaalChIP in Ref. 19 and performed a stage-by-stage analysis of the underlying SNP set (see **Fig. 3D** and **line 275-290**). It turned out that the fraction of BaalChIP ASB SNPs recovered by ADAstra was different for different cell types, with most of ASBs recovered for the cell types with the deepest sequencing coverage. On the other hand, we did not observe any particular stage critically inducing disagreement between ADAstra and BaalChIP pipelines. On the contrary, for all cell types, the number of BaalChIP ASB SNP recovered by ADAstra monotonously decreased from one SNP filtering stage to another, suggesting that there was no particular bottleneck defining the sensitivity of the ADAstra pipeline. As more sites were recovered for cell types with better coverage, one can predict that the difference between different ASB calling pipelines would decrease as soon as more experimental data sets are obtained for analysis.

The authors infer BAD under the assumption that the majority of SNVs in a region are independent of each other, saying that an ASB event in one SNV does not imply ASB somewhere nearby. However, given the increasing body of literature that attributes an important role to TFs and other proteins in steering local genome organization by aggregation, this assumption does not necessarily hold. Where TF binding shows such coordinated effects, BADs might thus be wrongly inferred, resulting in a substantial portion of false negatives.

Indeed, but the alternative would be tantamount to penalizing short high-BAD segments and could lead to an increased false-positive rate instead. In ADAstra we took a conservative approach, and (as mentioned above) this allowed us to capture most of the robust ASBs. We have added a note in the Discussion (see **line 523-535**).

Indeed, the authors list that their ASB calling includes “only” 38 – 63% of ASBs reported elsewhere. However, it is not obvious from the text why ADAstra is missing up to 60% of ASBs from existing sources, especially since one can assume that most ASBs that are inferred from smaller datasets should be included in the total set of

ASB calls from a larger resource. One major cause for this high number of false negatives could be the reason detailed above, but there may be other ones, all of which should be carefully investigated, as they may undermine the conceptual robustness of the presented workflow.

It is not trivial to directly compare ASB sets derived from different ChIP-Seq data. Indeed, the limited overlap between ADAstra and other collections that we showed e.g. in Supplementary Fig. S5 was rather puzzling. To clarify the reliability of our workflow, we estimated an overlap between ADAstra and the 'robust' ASB set consisting of 2039 SNPs with ASBs detected in any two of four pre-existing ASB sets (BaalChIP, Shi et al., Cavalli et al., AlleleDB). ADAstra ASBs cover 1573 (77%) of 2039 SNPs of the robust set, see **line 267-274** in the revised version of the manuscript. Of note, these four existing ASB data sets also poorly overlap each other (see newly introduced **Supplementary Table 4**), suggesting the unique ASBs are either specific to a particular ChIP-Seq data set or require pipeline-specific sensitivity improvements to be captured.

Do the authors suggest that the ASB distance distribution is bimodal with the shorter distance falling within the typical ChIP-seq peak width because more reads fall in that region and thus improve statistical power? The reason for observing such a bimodal distribution is not obvious to the reader and should be clarified.

This is related not only to the improved statistical significance of ASB calling but also to the mere presence of detectable SNPs passing coverage thresholds. We have rephrased an existing statement in the manuscript (**line 263-365**) to make it clearer.

Why did the authors use the ratio of log p-values for their motif analysis instead of using the relative affinities as obtained directly from the motif scores?

The log-odds scores of position weight matrices are very efficient in discriminative modeling of binding sites (i.e. determining bound and non-bound sites), but their relation to binding energy is not biologically relevant under saturation conditions (see a biophysical model in doi:10.1101/gr.1271603). In this setting, raw scores become not only non-relevant in the biological sense, but also technically non-comparable between different position weight matrices, while motif P-values provide a uniform way of assessment of the binding specificity. Considering analysis of regulatory SNPs, a deeper discussion of the motif annotation is given in Methods in [doi:10.1186/s12864-016-2728-9], see also Introduction in [doi:10.1093/bioinformatics/btq378].

The general concordance of motif presence/absence and ASB is encouraging and ADAstra should thus provide a valuable resource for further, more in-depth studies. In this regard, it would be interesting to see if there are any examples of TFs that have sufficient motif variations overlapping ASBs that a semi-quantitative comparison of ASB against motif score would be possible?

Indeed, huge ASB data allows such comparison (e.g. in a way similar to that used for plotting **Figure 4A**). However, the results will likely depend on the reliability of particular position weight matrices, so we consider robust ASB data as an opportunity for quantitative benchmarking of specificity models of transcription factor binding, see e.g. doi:10.1101/253427 and **line 540-543**.

From the main text, it remains unclear what exactly the two binary classification problems for machine learning are. The authors should clarify this section.

Indeed, there is a single classification problem (ASBs vs non-ASBs), but two subtasks depending on whether a general assessment is performed (ASB in any of cell types/for any of TFs) or if a model employs TF- or cell-type specificity. We have rewritten the respective section to improve clarity, see **line 368** in the revised manuscript.

Besides, a biological motivation why specific features are chosen would greatly help the reader to better understand the section. For instance, why are p-values instead of direct TF motif scores used (see also above)?

For decision trees, a monotonous transformation does not significantly affect the outcome (and, for a particular TF, P-values are monotonously dependent on scores). For multi-TF analysis, P-values might provide a better uniform scaling (see above).

In particular, a summary of the importance of the individual features would justify why this section is in the paper.

Indeed, we missed an opportunity to include a proper overview of feature importance estimates, which facilitates the interpretation of results. We have updated the section as suggested, see the updated **Supplementary Fig. S8** and **line 387-397** in the manuscript.

ATAC-seq differential accessibility is expected to correlate with TF ASB, given that for many TFs the majority of peaks fall within accessible regions. Thus, differentiating between TFs that are capable/not capable of binding to inaccessible regions could provide useful information on the importance of TF binding affinity to their cognate sites.

We used accessibility data from Ref 69 which does not represent the diversity of cell types and conditions used for the generation of the TF ChIP-Seq data. Thus, while these data being useful for machine learning per se, we would avoid using them to draw strong conclusions on chromatin preferences of particular TFs. However, by applying the proposed ML framework with a better matching DNase/ATAC-Seq data such analysis would be possible.

How many ASBs are expected to be missed because of the filtering procedures?

This question is linked to the previous one regarding the general robustness of the pipeline, please see the response above and the newly introduced panel **D** in **Fig. 3**.

How many cell lines are in fact majorly affected by aneuploidy?

Commonly used cell types are often aneuploid. Particularly, K562 is mostly triploid, MCF7 is mostly triploid, and 59 of 121 cell types overlapping between ADASTRA and COSMIC also have a median copy number above 2. We have added a note in Discussion, **line 523-527**.

***How many new ASB events were identified due to the suggested BAD approach?
Does one find many more ASB events compared to the already existing approaches?***

As the reviewer correctly pointed out, in comparison to simple binomial scoring, the BAD approach should result in more false negatives, i.e. it prioritizes specificity over sensitivity. At the same time, it should be more sensitive than using beta-binomial modeling that could over-inflate dispersion estimates. A quantitative analysis of 'why a particular ASB was missed by another pipeline' is hindered due to intrinsically different read mapping, ASB scoring, and P-value aggregation schemes used in different pipelines (see the newly introduced Supplementary Table 4 for pairwise comparison of selected ASB collections). We did our best to test the reliability of our approach, which recovers most of the robustly detected ASBs and hits significant eQTLs with the same frequency or better compared to the published approaches (see the responses above and, particularly, new panel **C** on **Fig. 5**, updated **Supplementary Fig. S9**, and newly added panel **D** on **Fig. 3**).

In the section “Motif annotation is concordant with ASB calls” the authors state that for “~10-20% of SNVs more reads are attracted to the weaker motif hit”. It would be nice to get some additional insights here and identify TFs that share the same ABSs, for instance.

We have performed a general analysis (see **Supplementary Fig. S6** and **line 296**) and found many cases of inter-TF ASBs that might be responsible for motif-discordant or no-motif-hit ASBs. A more detailed analysis would probably require a separate focused publication.

Additionally, it would be interesting to see how many TFs are there with a clear motif delta but with no ASBs.

It is an interesting suggestion but in fact, we did not observe such cases for motifs of TFs for which there were enough ChIP-Seq data. This might be a disadvantage of ADASTRA framework, in which it is not possible to reliably identify non-ASB sites. Particularly, as it was pointed out, we prefer false negatives to false positives, and thus cell-type or TF-specific ASB might be underscored, sometimes due to statistical aggregation of P-values, sometimes

due to aggressive BAD correction. The fraction of motif-concordant ASBs remains high even at higher FDR levels (see **Figure 4B**).

The description of the staveplot remains unclear and should be clarified. Therefore, the statement about less conserved positions allowing for more options is arguable, especially considering sites neighboring to position 6.

The most conserved motif positions prefer to carry ASBs involving the major nucleotide of the motif (i.e. pairs of <Major,Any> alleles, see positions 3-7-9-10 in Figure 4D with mostly unicolor beads on the strings), while less conserved positions allow <Any,Any> allele pairs (with beads of multiple colors). We have included more details in the respective revised section of the manuscript (**line 348-352**).

Response to reviewers' comments - Reviewer #2

On line 126, the authors say that they filtered the variant calls by "low-covered variants". I could not find how they define low-covered variants. I think it is important for the reader to know what precise definition was used here.

Indeed, ADAstra has two different thresholds for coverage (for SNPs used during BAD calling, and for candidate ASB SNPs). We have explicitly clarified the respective statements in the manuscript (see **line 126** and **line 404**).

On line 139, the authors indicate that "3000 ASBs for cell types passing an adjusted p-value of 0.05". How was this adjustment made? Was this a strict Bonferroni adjustment? Or some other type of multiple testing adjustment?

In the revised version of the manuscript, we have explicitly stated the usage of Benjamini-Hochberg (FDR) correction (**line 139**).

On line 576, there is an equation where there are only rectangles in both the numerator and denominator.

Indeed, the parameters of the binomial coefficient were empty due to the technical error on our side. We have updated the equation in the revised version of the manuscript (**line 655**).

REVIEWERS' COMMENTS

Reviewer #1 (Remarks to the Author):

In the rebuttal, Abramov et al. aimed to comprehensively address the questions of the reviewers and amended the main text together with supplementary material. Despite rather detailed explanations, however, the answers did not manage to significantly improve our understanding of the utilized methodology in a biologically relevant context, as was pointed out in the first review. In addition, several answers appear rather dismissive and only tangentially address raised issues. For instance, the question about identification of TFs sharing the same ABSs, and the one about the qualitative evaluation of TFs with a clear motif delta but no ASBs, were both addressed only in very general terms. Furthermore, the answer to the question of overlap estimation between the ADAstra database and other resources is also not sufficiently detailed, as the validity of the described approach to the 'robust' set identification is not justified and the underlying biological intuition is not provided. Finally, the authors added an explanation to the stove plot section that reads as a figure legend rather than a motivation. The latter would spell out for the reader how binding of a TF is expected to respond to DNA variation that impacts either conserved (key amino acid contacts) or less conserved positions in the PWM. Despite the technical explanation, it still remains a mystery to us why the ASB effect size is not higher at conserved positions, as would be expected from the binding model.

Thus, despite the clear merits of this work, which consist of assembling the ADAstra database by means of a new computational approach that identifies ASB events even for TFs and cell lines that would otherwise lack statistical power to do so, the study, in its current form, is unlikely to appeal to a broad readership as one would normally expect for a Nature Communications paper. Indeed, as it is written now, this work seems better suited for more specialized journals in computational biology / bio-informatics such as PLoS Computational Biology, BMC Bioinformatics, etc.

To conclude, Abramov et al. placed great emphasis on the computational aspects of their study but did not manage to properly contextualize the biological implications of their results and to implement a narrative that makes their findings accessible and understandable to a broader audience.

Reviewer #3 (Remarks to the Author):

Overall, I enjoyed reading this manuscript by Abramov et al. on identification of ASBs while considering background allelic dosage. While I agree with the comments by Reviewer #1 about the importance of benchmarking new methods, I believe the authors have reasonably demonstrated that their method is able to recover biologically relevant ASB events from large numbers of ChIP-seq data. A key consideration, in my opinion, is the fact that the ADAstra pipeline does not require gDNA information to identify CNVs. In contrast, methods such as BaalChIP rely on matching gDNA information from each sample to calculate background allele frequencies. Therefore, ADAstra is applicable to a substantially larger number of datasets. In other words, in addition to the sensitivity and specificity of different methods when applied to the same data, in my opinion we should also consider the scope of applicability of those methods when evaluating their merits. Figure 5C suggests that the ADAstra database increases the number of ASB events by almost two orders of magnitude while maintaining the same level of enrichment of known eQTLs as BaalChIP. The authors have also performed detailed analyses to explore why ADAstra ASBs only partially overlap those of other datasets, and to address other concerns such as concordance between ASBs and TF motifs. Overall, I believe this paper presents a resource that will likely be used by many researchers, and the authors have reasonably demonstrated its reliability and applicability.

I have a few minor comments:

- Line 283: "[...] we did not observe any particular stage critically inducing disagreement between ADAstra and BaalChIP pipelines [...]". However, Figure 3D clearly shows that the "basic coverage

filters" have a disproportionately large effect on SNP numbers, with a majority of BaalChIP SNPs in many cell lines removed at the stage of filtering low-coverage SNPs. Can the authors please revise this paragraph to better reflect the observations?

- Line 329: "Yet, for ~10-20% of SNVs, the motif hit odds-ratios (corrected for BAD) are discordant with the ASB disbalance [...]". How was the motif hit odds-ratio "corrected for BAD"? I presumed the motif hit scores (and therefore their odds-ratios) should not depend on BAD or any factor other than the genomic sequence. Did the authors mean "[...] the motif hit odds-ratios are discordant with the ASB disbalance (corrected for BAD) [...]?"

RESPONSE TO REVIEWERS' COMMENTS

Reviewer #1 (Remarks to the Author)

In the rebuttal, Abramov et al. aimed to comprehensively address the questions of the reviewers and amended the main text together with supplementary material. Despite rather detailed explanations, however, the answers did not manage to significantly improve our understanding of the utilized methodology in a biologically relevant context, as was pointed out in the first review. In addition, several answers appear rather dismissive and only tangentially address raised issues. For instance, the question about identification of TFs sharing the same ASBs, ...

We performed a basic analysis of TFs preferably sharing the same ASBs (see Supplementary Figure S6). We were able to identify many TF pairs with a statistically significant number of shared ASB (with FDR-corrected values of Fisher's exact test up to 10^{-300}). On the one hand, we were unable to formulate an apparent interpretation of the results, for instance, in terms of known protein-protein interactions according to STRING-DB data (as shown in Supplementary Figure S6). The most apparent overlap was found between ASBs of chromatin-interacting epigenetic factors and related proteins, suggesting that many of these events occur in regions of allele-specific chromatin accessibility and can be explained by the chromosome accessibility bias as a confounding factor. On the other hand, there were strong known cases reflecting protein complexes, as pointed out in Results (see lines 297-308 of the revised manuscript). An in-depth analysis should be indeed possible with ADAstra data but we consider it out of the scope of the current manuscript.

... and the one about the qualitative evaluation of TFs with a clear motif delta but no ASBs, were both addressed only in very general terms.

In regard to TFs with clear motif delta but no ASBs, first, we did not observe any TFs with sufficiently represented ASBs with negative or zero correlation of the ChIP-Seq allelic ratio with the motif delta (See Figure 4B, top panel, "Fraction of concordant SNPs"). Yet, almost for any TFs, there were "discordant" SNPs with the ChIP-Seq allelic imbalance disagreeing with the motif prediction scores. We assume those cases receive significant contribution either from chromatin accessibility or from imperfect motif models. Comprehensive analysis of allele-specific chromatin accessibility or validating the reliability of motif models of particular TFs requires dedicated studies (see e.g. Ref.33 and Ref.16 of the manuscript). We have revised the manuscript (see lines 312-320 and 336-340) to include more details directly in the text.

Furthermore, the answer to the question of overlap estimation between the ADAstra database and other resources is also not sufficiently detailed, as the validity of the described approach to the 'robust' set identification is not justified and the underlying biological intuition is not provided.

The rationale behind 'robust' set of ASBs is very straightforward: if an ASB is recognized by more than one approach or found in more than one collection then it is more likely to be the true event (i.e. reproducible between experiments and/or computational pipelines). Ensemble or voting methods are commonly used in various areas of bioinformatics so we did not consider including an explicit substantiation. For clarity, we have changed 'robust' to 'reproducible' in the revised version of the manuscript.

Finally, the authors added an explanation to the stove plot section that reads as a figure legend rather than a motivation. The latter would spell out for the reader how binding of a TF is expected to respond to DNA variation that impacts either conserved (key amino acid contacts) or less conserved positions in the PWM. Despite the technical explanation, it still remains a mystery to us why the ASB effect size is not higher at conserved positions, as would be expected from the binding model.

In fact, the ASB effect size is higher at conserved positions. This effect is exhibited to a varying degree (due to different motif lengths and different number of ASBs per position, see Supplementary Figure S7), but it is very clear for CEBPB as shown in Figure 4d. We have rewritten the respective paragraph to better explain the stoveplot and explicitly highlight this observation (line 370-370).

Thus, despite the clear merits of this work, which consist of assembling the ADAstra database by means of a new computational approach that identifies ASB events even for TFs and cell lines that would otherwise lack statistical power to do so, the study, in its current form, is unlikely to appeal to a broad readership as one would normally expect for a Nature Communications paper. Indeed, as it is written now, this work seems better suited for more specialized journals in computational biology / bioinformatics such as PLoS Computational Biology, BMC Bioinformatics, etc.

We would like to highlight that in this manuscript we did not only present a novel framework suitable for systematic large-scale analysis of diverse datasets but also provided several biologically relevant applications and insights. Systematic assessment of the ChIP-Seq signal across experiments provides a valuable source of information on TF binding affinity dependence on the DNA sequence, and can give insights into molecular mechanisms of gene regulation and genotype-phenotype associations (see the last section of Results, starting from page 14), especially when supported by eQTL data. In this setting, we fully believe the paper and the accompanying database would be interesting to a broad readership, from technical bioinformaticians to clinical geneticists.

To conclude, Abramov et al. placed great emphasis on the computational aspects of their study but did not manage to properly contextualize the biological implications of their results and to implement a narrative that makes their findings accessible and understandable to a broader audience.

We agree that the presented methodology is based on a complex statistical framework which might be too technical for a broader audience. However, we believe that the results of our software pipeline reveal the global landscape of allele-specific binding in the human genome at a yet inaccessible scale and can be important for a large number of new studies from the biology of transcription initiation to medical genetics.

Reviewer #3 (Remarks to the Author)

Overall, I enjoyed reading this manuscript by Abramov et al. on identification of ASBs while considering background allelic dosage. While I agree with the comments by Reviewer #1 about the importance of benchmarking new methods, I believe the authors have reasonably demonstrated that their method is able to recover biologically relevant ASB events from large numbers of ChIP-seq data. A key

consideration, in my opinion, is the fact that the ADAstra pipeline does not require gDNA information to identify CNVs. In contrast, methods such as BaalChIP rely on matching gDNA information from each sample to calculate background allele frequencies. Therefore, ADAstra is applicable to a substantially larger number of datasets. In other words, in addition to the sensitivity and specificity of different methods when applied to the same data, in my opinion we should also consider the scope of applicability of those methods when evaluating their merits. Figure 5C suggests that the ADAstra database increases the number of ASB events by almost two orders of magnitude while maintaining the same level of enrichment of known eQTLs as BaalChIP. The authors have also performed detailed analyses to explore why ADAstra ASBs only partially overlap those of other datasets, and to address other concerns such as concordance between ASBs and TF motifs. Overall, I believe this paper presents a resource that will likely be used by many researchers, and the authors have reasonably demonstrated its reliability and applicability.

We deeply thank the respectable reviewer for the favorable assessment of our work.

I have a few minor comments:

- Line 283: “[...] we did not observe any particular stage critically inducing disagreement between ADAstra and BaalChIP pipelines [...]”. However, Figure 3D clearly shows that the “basic coverage filters” have a disproportionately large effect on SNP numbers, with a majority of BaalChIP SNPs in many cell lines removed at the stage of filtering low-coverage SNPs. Can the authors please revise this paragraph to better reflect the observations?

We fully agree and have revised the text accordingly (lines 284-289).

- Line 329: “Yet, for ~10-20% of SNVs, the motif hit odds-ratios (corrected for BAD) are discordant with the ASB disbalance [...]”. How was the motif hit odds-ratio “corrected for BAD”? I presumed the motif hit scores (and therefore their odds-ratios) should not depend on BAD or any factor other than the genomic sequence. Did the authors mean “[...] the motif hit odds-ratios are discordant with the ASB disbalance (corrected for BAD) [...]”?

Indeed, the motif hit odds-ratios are not dependent on BAD. We apologize for this confusing statement. We have corrected the text as suggested.